# Scalable Neural Network Verification with Branch-and-bound Inferred Cutting Planes

**Duo Zhou**[1]    **Christopher Brix**[2]    **Grani A Hanasusanto**[1]    **Huan Zhang**[1]

[1]University of Illinois Urbana-Champaign    [2]RWTH Aachen University

{duozhou2,gah}@illinois.edu   brix@cs.rwth-aachen.de   huan@huan-zhang.com

## Abstract

Recently, cutting-plane methods such as GCP-CROWN have been explored to enhance neural network verifiers and made significant advances. However, GCP-CROWN currently relies on *generic* cutting planes ("cuts") generated from external mixed integer programming (MIP) solvers. Due to the poor scalability of MIP solvers, large neural networks cannot benefit from these cutting planes. In this paper, we exploit the structure of the neural network verification problem to generate efficient and scalable cutting planes *specific* for this problem setting. We propose a novel approach, Branch-and-bound Inferred Cuts with COnstraint Strengthening (BICCOS), which leverages the logical relationships of neurons within verified subproblems in the branch-and-bound search tree, and we introduce cuts that preclude these relationships in other subproblems. We develop a mechanism that assigns influence scores to neurons in each path to allow the strengthening of these cuts. Furthermore, we design a multi-tree search technique to identify more cuts, effectively narrowing the search space and accelerating the BaB algorithm. Our results demonstrate that BICCOS can generate hundreds of useful cuts during the branch-and-bound process and consistently increase the number of verifiable instances compared to other state-of-the-art neural network verifiers on a wide range of benchmarks, including large networks that previous cutting plane methods could not scale to. BICCOS is part of the $\alpha,\beta$-`CROWN` verifier, **the VNN-COMP 2024 winner**. The code is available at https://github.com/Lemutisme/BICCOS.

## 1   Introduction

Formal verification of neural networks (NNs) has emerged as a critical challenge in the field of artificial intelligence. In canonical settings, the verification procedure aims to prove certified bounds on the outputs of a neural network given a specification, such as robustness against adversarial perturbations or adherence to safety constraints. As these networks become increasingly complex and are deployed in sensitive domains, rigorously ensuring their safety and reliability is paramount. Recent research progress on neural network verification has enabled safety or performance guarantees in several mission-critical applications [55, 53, 51, 13, 60, 57].

The branch and bound (BaB) procedure has shown great promise as a verification approach [12, 59, 21]. In particular, the commonly used ReLU activation function exhibits a piecewise linear behavior, and this property makes ReLU networks especially suitable for branch and bound techniques. During the branching process, one ReLU neuron is selected and two subproblems are created. Each subproblem refers to one of the two ReLU states (inactive or active) and the bounds of the neurons' activation value are tightened correspondingly. BaB systematically branches ReLU neurons, creates a search tree of subproblems, and prunes away subproblems whose bounds are tight enough to guarantee the verification specifications. Although our work focuses on studying the canonical ReLU settings, branch-and-bound has also demonstrated its power in non-ReLU cases [48]. Despite its success in many tasks, the efficiency and effectiveness of the branch-and-bound procedure heavily depend on the number of subproblems that can be pruned.

Recent work has explored the use of cutting plane methods in neural network verification. *Cutting planes* ("cuts") are additional (linear) constraints in the optimization problem for NN verification to tighten the bound without affecting soundness. By introducing carefully chosen cuts, they can significantly tighten the bounds in each subproblem during BaB, leading to more pruned subproblems and faster verification. However, it is quite challenging to generate effective cutting planes for large-scale NN verification problems. For example, GCP-CROWN [61] relied on strong cutting planes generated by a mixed integer programming (MIP) solver, which includes traditional *generic* cutting planes such as the Gomory cuts [24], but its effectiveness is limited to only small neural networks. The key to unlocking the full potential of BaB lies in the development of scalable cutting planes that are *specific* to the NN verification problem and also achieve high effectiveness.

In this paper, we propose a novel methodology, Branch-and-bound Inferred Cuts with COnstraint Strengthening (BICCOS), that can produce effective and scalable cutting planes specifically for the NN verification problem. Our approach leverages the information gathered during the branch-and-bound process to generate cuts on the fly. First, we show that by leveraging any verified subproblems (e.g., a subproblem is verified when neurons A and B are both split to active cases), we can deduce cuts that preclude certain combinations of ReLU states (e.g., neurons A and B cannot be both active). This cut may tighten the bounds for subproblems in other parts of the search tree, even when neurons A and B have not yet been split. Second, to make these cuts effective during the regular BaB process, we show that they can be strengthened by reducing the number of branched neurons in a verified subproblem (e.g., we may conclude that setting neuron B to active is sufficient to verify the problem, drop the dependency on A, and remove one variable from the cut). BICCOS can find and strengthen cuts on the fly during the BaB process, adapting to the specific characteristics of each verification instance. Third, we propose a pre-solving strategy called "multi-tree search" which proactively looks for effective cutting planes in many shallow search trees before the main BaB phase starts. Our main contributions can be summarized as follows:

- We first identify the opportunity of extracting effective cutting planes during the branch-and-bound process in NN verification. These cuts are *specific* to the NN verification setting and *scalable* to large NNs that previous state-of-the-art cutting plane methods such as GCP-CROWN with MIP-cuts cannot handle. Our cuts can be plugged into existing BaB-based verifiers to enhance their performance.

- We discuss several novel methods to strengthen the cuts and also find more effective cuts. Strengthening the cuts is essential for finding effective cuts on the fly during the regular BaB process that can tighten the bounds in the remaining domains. Performing a multi-tree search allows us to identify additional cuts before the regular BaB process begins.

- We conduct empirical comparisons against many verifiers on a wide range of benchmarks and consistently outperform state-of-the-art baselines in VNN-COMP [4, 40, 10]. Notably, we can solve the very large models like those for `cifar100` and `tinyimagenet` benchmarks and outperform all state-of-the-art tools, to which GCP-CROWN with MIP-based cuts cannot scale.

## 2 Background

**Notations** Bold symbols such as $\boldsymbol{x}^{(i)}$ denote vectors; regular symbols such as $x_j^{(i)}$ indicate scalar components of these vectors. $[N]$ represents the set $\{1, \ldots, N\}$, and $\mathbf{W}_{:,j}^{(i)}$ is the $j$-th column of the matrix $\mathbf{W}^{(i)}$. A calligraphic font $\mathcal{X}$ denotes a set and $\mathbb{R}^n$ represents the $n$ dimensional real number space.

**The NN Verification Problem** An $L$-layer ReLU-based DNN can be formulated as $f : \boldsymbol{x} \mapsto \boldsymbol{x}^{(L)}$, s.t. $\{\boldsymbol{x}^{(i)} = \boldsymbol{W}^{(i)}\hat{\boldsymbol{x}}^{(i-1)} + \boldsymbol{b}^{(i)}, \hat{\boldsymbol{x}}^{(i)} = \sigma(\boldsymbol{x}^{(i)}), \boldsymbol{x}^{(0)} = \boldsymbol{x}; \quad i \in [L]\}$, where $\sigma$ represents the ReLU activation function, with input $\boldsymbol{x} =: \hat{\boldsymbol{x}}^{(0)} \in \mathbb{R}^{d^{(0)}}$ and the neuron network parameters weight matrix $\boldsymbol{W}^{(i)} \in \mathbb{R}^{d^{(i)} \times d^{(i-1)}}$ and bias vector $\boldsymbol{b}^{(i)} \in \mathbb{R}^{d^{(i)}}$ for each layer $i$. This model sequentially processes the input $\boldsymbol{x}$ through each layer by computing linear transformations followed by ReLU activations. The scalar values $\hat{x}_j^{(i)}$ and $x_j^{(i)}$ represent the post-activation and pre-activation values of the $j$-th neuron in the $i$-th layer, respectively. We write $\hat{f}_j^{(i)}(\boldsymbol{x})$ and $f_j^{(i)}(\boldsymbol{x})$ for $\hat{x}_j^{(i)}$ and $x_j^{(i)}$, respectively, when they depend on a specific $\boldsymbol{x}$.

In practical applications, the input $\boldsymbol{x}$ is confined within a perturbation set $\mathcal{X}$, often defined as an $\ell_p$ norm ball. The verification task involves ensuring a specified output property for any $\boldsymbol{x} \in \mathcal{X}$. For example, it may be required to verify that the logit corresponding to the true label $f_i(\boldsymbol{x})$ is consistently higher than the logit for any other label $f_j(\boldsymbol{x})$, thus ensuring $f_i(\boldsymbol{x}) - f_j(\boldsymbol{x}) > 0, \quad \forall j \neq i$. By

integrating the verification specification as an additional layer with only one output neuron, we define the canonical verification problem as an optimization problem:

$$f^\star = \min_{\boldsymbol{x} \in \mathcal{X}} f(\boldsymbol{x}), \tag{1}$$

where the optimal value $f^\star \geq 0$ confirms the verifiable property. We typically use the $\ell_\infty$ norm to define $\mathcal{X} := \{\boldsymbol{x} : \|\boldsymbol{x} - \boldsymbol{x}_0\|_\infty \leq \epsilon\}$, with $\boldsymbol{x}_0$ as a baseline input. However, extensions to other norms and conditions are also possible [44, 58]. When we compute a lower bound $\underline{f}^\star \leq f^\star$, we label the problem **UNSAT** if the property was verified and $\underline{f}^\star \geq 0$, i.e. the problem of finding a concrete input that violates the property is **unsatisfiable**, or in other words **infeasible**. If $\underline{f}^\star < 0$, it is unclear whether the property might hold or not. We refer to this as **unknown**.

**MIP Formulation and LP Relaxation**  In the optimization problem (1), a non-negative $f^\star$ indicates that the network can be verified. However, due to the non-linearity of ReLU neurons, this problem is non-convex, and ReLU neurons are typically relaxed with linear constraints to obtain a lower bound for $f^\star$. One possible solution is to encode the verification problem using Mixed Integer Programming (MIP), which encodes the entire network architecture, using $\boldsymbol{x} \in \mathbb{R}^{d^{(i)}}$ to denote the **pre-activation neurons**, $\hat{\boldsymbol{x}} \in \mathbb{R}^{d^{(i)}}$ to denote **post-activation neurons** for each layer, and **binary ReLU indicator** $z_j^{(i)} \in \{0,1\}$ to denote **(in)active neuron** for each unstable neuron. A lower bound can be computed by letting $z_j^{(i)} \in [0,1]$ and therefore relaxing the problem to an LP formulation. There is also an equivalent Planet relaxation [20]. We provide the detailed definition in Appendix A. In practice, this approach is computationally too expensive to be scalable.

**Branch-and-bound**  Instead of solving the expensive LP for each neuron, most existing NN verifiers use cheaper methods such as abstract interpretation [49, 23] or bound propagation [16, 62, 54, 59] due to their efficiency and scalability. However, because of those relaxations, the lower bound for $f^\star$ might eventually become too weak to prove $f^\star \geq 0$. To overcome this issue, additional constraints need to be added to the optimization, without sacrificing soundness.

The branch-and-bound (BaB) framework, illustrated in Figure 1, is a powerful approach for neural network verification that many state-of-the-art verifiers are based on [12, 54, 16, 29, 21]. BaB systematically tightens the lower bound of $f^\star$ by splitting unstable ReLU neurons into two cases: $x_j \geq 0$ and $x_j \leq 0$ (branching step), which defines two subproblems with additional constraints. In each subproblem, neuron $x_j$ does not need to be relaxed, leading to a tighter lower bound (bounding step). Note that $\{x_j^{(i)} \leq u_j^{(i)} = 0\}$ and $\{x_j^{(i)} \geq l_j^{(i)} = 0\}$ in the Planet relaxation are **equivalent** to $\{z_j^{(i)} = 0\}$ and $\{z_j^{(i)} = 1\}$ in the LP relaxation, respectively, see Lemma A.1 in Appendix A. Subproblems with a positive lower bound are successfully verified, and no further splitting is required. The process repeats on subproblems with negative lower bounds until all unstable neurons are split, or all subproblems are verified. If there are still domains with negative bounds after splitting all unstable neurons, a counter-example can be constructed.

Figure 1: Each node represents a subproblem in the BaB process by splitting unstable ReLU neurons. Green nodes indicate paths that have been verified and pruned, while blue nodes represent domains that are still unknown and require further branching.

**General Cutting Planes (GCP) in NN verification**  A (linear) cutting plane ("cut") is a linear inequality that can be added to a MIP problem, which does not eliminate any feasible integer solutions but will tighten the LP relaxation of this MIP. For more details on cutting plane methods, we refer readers to the literature on integer programming [7, 14]. In the NN verification setting, it may involve variables $\boldsymbol{x}^{(i)}$ (pre-activation), $\hat{\boldsymbol{x}}^{(i)}$ (post-activation) from any layer $i$, and $\boldsymbol{z}^{(i)}$ (binary ReLU indicators) from any unstable neuron. Given $N$ cutting planes in matrix form as [61]:

$$\sum_{i=1}^{L-1} \left( \boldsymbol{H}^{(i)} \boldsymbol{x}^{(i)} + \boldsymbol{G}^{(i)} \hat{\boldsymbol{x}}^{(i)} + \boldsymbol{Q}^{(i)} \boldsymbol{z}^{(i)} \right) \leq \mathbf{d} \tag{2}$$

where $\boldsymbol{H}^{(i)}$, $\boldsymbol{G}^{(i)}$, and $\boldsymbol{Q}^{(i)}$ are the matrix coefficients corresponding to the cutting planes. Based on this formulation, one can introduce arbitrary valid cuts to tighten the relaxation. GCP-CROWN [61]

allows us to lower bound (1) with arbitrary linear constraints in (2) using GPU-accelerated bound propagation, without relying on an LP solver.

In [61], the authors propose to find new cutting planes by employing an MIP solver. By encoding (1) as a MIP problem, the MIP solver will identify potentially useful cutting planes. Usually, these cuts would be used by the solver itself to solve the MIP problem. Instead, [61] applied these cuts using GPU-accelerated bound propagation. This approach shows great improvements on many verification problems due to the powerful cuts, but it depends on the ability of the MIP solver to identify relevant cuts. As the networks that are analyzed increase in size, the respective MIP problem increases in complexity, and the MIP solver may not return any cuts before the timeout is reached. In the next chapter, we will describe a novel approach to generate effective and scalable cutting planes.

## 3 Branch-and-bound Inferred Cuts with Constraint Strengthening (BICCOS)

### 3.1 Branch-and-bound Inferred Cuts

The first key observation in our algorithm is that the UNSAT subproblems in the BaB search tree include valuable information. If the lower bound of a subproblem leads this subproblem to be UNSAT (e.g., subproblems with green ticks in Fig. 1), it signifies that restricting the neurons along this path to the respective positive/negative regimes allows us to verify the property. A typical BaB algorithm would stop generating further subproblems at this juncture, and continue with splitting only those nodes that have not yet been verified. Crucially, no information from the verified branch is transferred to the unverified domains. However, sometimes, this information can help a lot.

*Example.* As shown in Fig. 1, assume that after splitting $x_1 \leq 0$ and $x_3 \leq 0$, the lower bound of this subproblem is found to be greater than $0$, indicating infeasibility. From this infeasibility, we can infer that the neurons $x_1$ and $x_3$ cannot simultaneously be in the inactive regime. To represent this relationship, we use the relaxed ReLU indicator variables $z_1, z_3 \in [0, 1]$ in the LP formulation and form the inequality $z_1 + z_3 \geq 1$. This inequality ensures that both $z_1$ and $z_3$ cannot be $0$ simultaneously. If we were to start BaB with a fresh search tree, this constraint has the potential to improve the bounds for *all* subproblems by tightening the relaxations and excluding infeasible regions.

We propose to encode the information gained from a verified subproblem as a new cutting plane. These cutting planes will be valid globally across the entire verification problem and all generated subproblems. We present the general case of this cutting plane below:

**Proposition 3.1.** *For a verified, or UNSAT, subproblem in a BaB search tree, let $\mathcal{Z}_+$ and $\mathcal{Z}_-$ be the set of neurons restricted to the positive and negative regimes respectively. These restrictions were introduced by the BaB process. Then, the BaB inferred cut can be formulated as:*

$$\sum_{i \in \mathcal{Z}_+} z_i - \sum_{i \in \mathcal{Z}_-} z_i \leq |\mathcal{Z}_+| - 1 \tag{3}$$

Proof deferred to Appendix B.1. The BaB inferred cut (3) will exclude the specific combination of positive and negative neurons that were proven to be infeasible in the respective branches. An example is shown in Fig. 2a.

Note that while similar cuts were explored in [9], they were not derived from UNSAT problems within the BaB process. In our framework, these cuts can theoretically be incorporated as cutting planes in the form of (2) and work using GCP-CROWN. However, a limitation exists: all elements in our cut are ReLU indicator $z$, although GCP-CROWN applies general cutting planes during the standard BaB process. It was not originally designed to handle branching decisions on ReLU indicator variables $z$ that may have (partially) been fixed already in previous BaB steps. When cuts are added, they remain part of the GCP-CROWN formulation. However, during the BaB process, some $z$ variables may be fixed to $0$ or $1$ due to branching, effectively turning them into constants This situation poses a challenge because the original GCP-CROWN formulation presented in [61] does not accommodate constraints involving these fixed $z$ variables, potentially leading to incorrect results. To address this issue, we need to extend GCP-CROWN to handle BaB on the ReLU indicators $z$, ensuring that the constraints and cuts remain valid even when some $z$ variables are fixed during branching.

**Extension of GCP-CROWN** To address this limitation, we propose an extended form of bound propagation for BaB inferred cuts that accommodates splitting on $z$ variables. In the original GCP-CROWN formulation, $\mathcal{I}^{(i)}$ represents the set of **initially** unstable neurons in layer $i$, for which $z$ cuts

were added due to their instability. However, during the branch-and-bound process, some of these neurons may be split, fixing their corresponding $z$ variables to 0 or 1, and thus their $z$ variables no longer exist in the original formulation. While updating all existing cuts by fixing these $z$ variables is possible, it is costly since the cuts for each subproblem must be fixed individually, as the neuron splits in each subproblem is different. Our contribution is to handle these split neurons by adding them to the splitting set $\mathcal{Z}$ in the new formulation below, without removing or modifying the existing cuts (all subdomains can still share the same set of cuts). This approach allows us to adjust the original bound propagation in [61, Theorem 3.1] to account for the fixed $z$ variables and their influence on the Lagrange dual problem, without altering the existing cuts. Suppose the splitting set for each layer $i$ is $\mathcal{Z}^{+(i)} \cup \mathcal{Z}^{-(i)} := \mathcal{Z}^{(i)} \subseteq \mathcal{I}^{(i)}$, and the full split set is $\mathcal{Z} = \bigcup_{i \in [L-1]} \mathcal{Z}^{(i)}$. The modifications to the original GCP-CROWN theorem are highlighted in brown in the following theorem.

**Theorem 3.2.** *[BaB Inferred Cuts Bound Propagation]. Given any BaB split set $\mathcal{Z}$, optimizable parameters $0 \leq \alpha_j^{(i)} \leq 1$ and $\beta, \mu, \tau \geq 0$, $\pi_j^{(i)\star}$ is a function of $Q_{:,j}^{(i)}$:*

$$g(\alpha, \beta, \mu, \tau) = -\epsilon \|\nu^{(1)\top} \mathbf{W}^{(1)} x_0\|_1 - \sum_{i=1}^{L} \nu^{(i)\top} \mathbf{b}^{(i)} - \beta^\top d + \sum_{i=1}^{L-1} \sum_{j \in \mathcal{I}^{(i)}} h_j^{(i)}(\beta)$$

*where variables $\nu^{(i)}$ are obtained by propagating $\nu^{(L)} = -1$ throughout all $i \in [L-1]$:*

$$\nu_j^{(i)} = \nu^{(i+1)\top} \mathbf{W}_{:,j}^{(i+1)} - \beta^\top (H_{:,j}^{(i)} + G_{:,j}^{(i)}), \ \ j \in \mathcal{I}^{+(i)}$$

$$\nu_j^{(i)} = \nu^{(i+1)\top} \mathbf{W}_{:,j}^{(i+1)} - \beta^\top (H_{:,j}^{(i)} + G_{:,j}^{(i)}) + \mu_j^{(i)}, \ \ j \in \mathcal{Z}^{+(i)},$$

$$\nu_j^{(i)} = -\beta^\top H_{:,j}^{(i)}, \ \ j \in \mathcal{I}^{-(i)},$$

$$\nu_j^{(i)} = -\beta^\top H_{:,j}^{(i)} - \tau_j^{(i)}, \ \ j \in \mathcal{Z}^{-(i)},$$

$$\nu_j^{(i)} = \pi_j^{(i)\star} + \alpha_j^{(i)} [\hat{\nu}_j^{(i)}]_- - \beta^\top H_{:,j}^{(i)}, \ \ j \in \mathcal{I}^{(i)} \setminus \mathcal{Z}^{(i)}$$

*Here, $\hat{\nu}_j^{(i)}$, $\pi_j^{(i)\star}$ and $\tau_j^{(i)}$ are defined for each initially unstable neuron that has not been split: $j \in \mathcal{I}^{(i)} \setminus \mathcal{Z}^{(i)}$, and $h_j^{(i)}(\beta)$ is defined for all unstable neurons $j \in \mathcal{I}^{(i)}$.*

$$[\hat{\nu}_j^{(i)}]_+ := \max(\nu^{(i+1)\top} \mathbf{W}_{:,j}^{(i+1)} - \beta^\top G_{:,j}^{(i)}, 0), \ \ [\hat{\nu}_j^{(i)}]_- := \min(\nu^{(i+1)\top} \mathbf{W}_{:,j}^{(i+1)} - \beta^\top G_{:,j}^{(i)}, 0)$$

$$\pi_j^{(i)\star} := \max \left( \min \left( \frac{u_j^{(i)} [\hat{\nu}_j^{(i)}]_+ - \beta^\top Q_{:,j}^{(i)}}{u_j^{(i)} - l_j^{(i)}}, [\hat{\nu}_j^{(i)}]_+ \right), 0 \right), \ j \in \mathcal{I}^{(i)} \setminus \mathcal{Z}^{(i)}$$

$$h_j^{(i)}(\beta) := \begin{cases} l_j^{(i)} \pi_j^{(i)\star} & \text{if } l_j^{(i)} [\hat{\nu}_j^{(i)}]_+ \leq \beta^\top Q_{:,j}^{(i)} \leq u_j^{(i)} [\hat{\nu}_j^{(i)}]_+ \text{ and } j \in \mathcal{I}^{(i)} \setminus \mathcal{Z}^{(i)}, \\ 0 & \text{if } \beta^\top Q_{:,j}^{(i)} \geq u_j^{(i)} [\hat{\nu}_j^{(i)}]_+ \text{ or } j \in \mathcal{Z}^{-(i)} \\ \beta^\top Q_{:,j}^{(i)} & \text{if } \beta^\top Q_{:,j}^{(i)} \leq l_j^{(i)} [\hat{\nu}_j^{(i)}]_+ \text{ or } j \in \mathcal{Z}^{+(i)} \end{cases}$$

Proof in Appendix B.2. The brown-highlighted modifications ensure that the bound propagation correctly accounts for the influence of the fixed ReLU indicator $z$ variables—resulting from branching decisions in the BaB process—on each layer's coefficients and biases. Notably, if we remove all cutting planes, this propagation method reduces to $\beta$-CROWN [54]. In this context, the new dual variables $\mu$ and $\tau$ correspond to the dual variable $\beta$ used for $x$ splits in $\beta$-CROWN. To handle splits, we only need to specify the sets $\mathcal{Z}^{+(i)}$ and $\mathcal{Z}^{-(i)}$ for each layer. By adjusting the dual variables and functions to reflect the fixed states of certain neurons, the extended bound propagation maintains tightness and correctness in the computed bounds.

## 3.2 Improving BaB Inferred Cuts via Constraint Strengthening and Multi-Tree Search

Theorem 3.2 allows us to use the cheap bound propagation method to search the BaB tree quickly to discover UNSAT paths to generate BaB inferred cuts by Proposition 3.1. However, our second key observation is that naively inferred cuts will not be beneficial in a regular BaB search process, illustrated in Fig. 2a, for example, all the subproblems after the split $x_1 \geq 0$ (all nodes on the right after the first split) imply $z_1 = 1$, and thus $z_1 + z_3 \geq 1$ always holds. Thus, we have to strengthen these cuts to make them more effective. We propose the Branch-and-bound Inferred Cut with COnstraint Strengthening (BICCOS) to solve this.

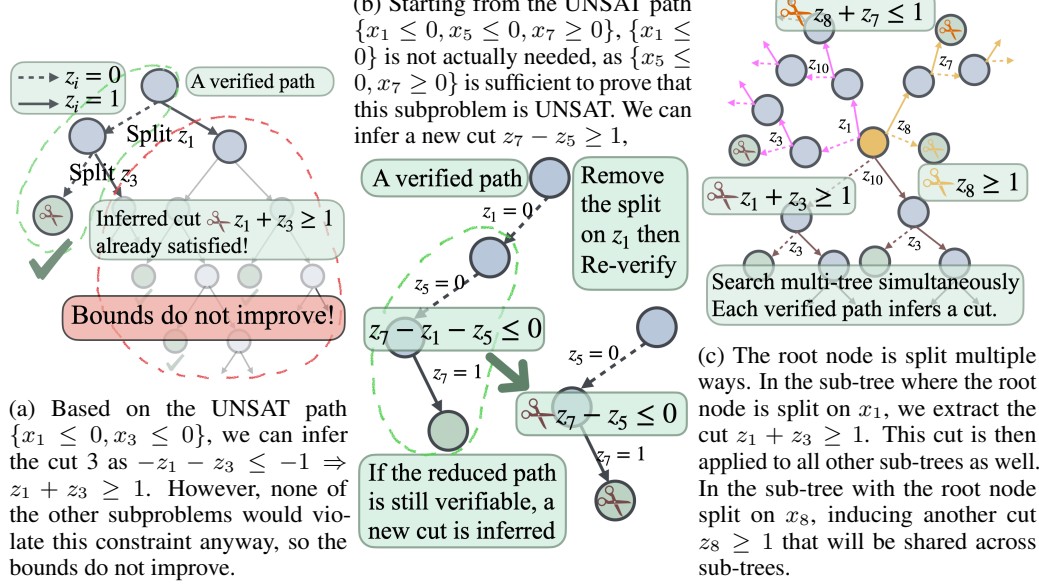

(a) Based on the UNSAT path $\{x_1 \le 0, x_3 \le 0\}$, we can infer the cut 3 as $-z_1 - z_3 \le -1 \Rightarrow z_1 + z_3 \ge 1$. However, none of the other subproblems would violate this constraint anyway, so the bounds do not improve.

(b) Starting from the UNSAT path $\{x_1 \le 0, x_5 \le 0, x_7 \ge 0\}$, $\{x_1 \le 0\}$ is not actually needed, as $\{x_5 \le 0, x_7 \ge 0\}$ is sufficient to prove that this subproblem is UNSAT. We can infer a new cut $z_7 - z_5 \ge 1$. If the reduced path is still verifiable, a new cut is inferred

(c) The root node is split multiple ways. In the sub-tree where the root node is split on $x_1$, we extract the cut $z_1 + z_3 \ge 1$. This cut is then applied to all other sub-trees as well. In the sub-tree with the root node split on $x_8$, inducing another cut $z_8 \ge 1$ that will be shared across sub-trees.

Figure 2: (2a): Inferred cut from UNSAT paths during BaB and why it fails in regular BaB. (2b): Constraint strengthening with Neuron Elimination Heuristic. (2c): Multi-tree search.

In a known UNSAT subproblem with multiple branched neurons, it is possible that a *subset* of the branching neurons is sufficient to prove UNSAT, shown in Fig. 2b. By focusing on this subset, we can strengthen the BaB inferred cuts by reducing the number of $z$ variables involved. From an optimization perspective, each cut corresponds to a hyperplane that separates feasible solutions from infeasible ones. Simplifying the cut by involving fewer $z$ variables reduces the dimensionality of the hyperplane, making it more effective at excluding infeasible regions without unnecessarily restricting the feasible space. Thus, we can draw a corollary:

**Corollary 3.3.** *Formally, consider the two cuts from Eq. (3):*

$$\sum_{i \in \mathcal{Z}_1^+} z_i - \sum_{i \in \mathcal{Z}_1^-} z_i \le |\mathcal{Z}_1^+| - 1 \tag{3.1}$$

$$\sum_{i \in \mathcal{Z}_2^+} z_i - \sum_{i \in \mathcal{Z}_2^-} z_i \le |\mathcal{Z}_2^+| - 1 \tag{3.2}$$

*If either $\mathcal{Z}_1^+ \subset \mathcal{Z}_2^+$ and $\mathcal{Z}_1^- \subseteq \mathcal{Z}_2^-$, or $\mathcal{Z}_1^+ \subseteq \mathcal{Z}_2^+$ and $\mathcal{Z}_1^- \subset \mathcal{Z}_2^-$, then any assignment $\{z_i\}$ satisfying the cut associated with the smaller split set $\mathcal{Z}$ also satisfies the other cut. This implies that the cut with the smaller set is strictly stronger than the one with the larger set.*

Proof deferred to Appendix B.3. This demonstrates that cuts with fewer variables can be more powerful because they operate in lower-dimensional spaces, allowing the hyperplane to more effectively position itself to exclude infeasible regions.

Now, the challenge is determining which unnecessary branches to remove to produce a cut with as few $z$ variables as possible. An important observation from Theorem 3.2 is that the dual variables $\mu$ and $\tau$, associated with the fixed ReLU indicators $z \in \mathcal{Z}^{+(i)}$ and $z \in \mathcal{Z}^{-(i)}$ respectively, reflect the impact of these fixed activations on the bound. Specifically, a positive dual variable $\mu > 0$ for some $z \in \mathcal{Z}^{+(i)}$ indicates that the fixed active state of these neurons contributes significantly to tightening the bound. Similarly, a positive $\tau > 0$ for some $z \in \mathcal{Z}^{-(i)}$ signifies that the fixed inactive neurons are influential in optimizing the bound. However, neurons with zero dual variables ($\mu = 0$ or $\tau = 0$) may not contribute to the current bound optimization. This observation suggests that these neurons might be candidates for removal to simplify the cut. But it's important to note that a zero dual variable does not guarantee that the corresponding neuron has no impact under all circumstances—it only indicates no contribution in the current optimization context. Simply removing all neurons with zero dual

variables might overlook their potential influence in other parts of the search space or under different parameter settings. Therefore, the challenge lies in deciding which neurons with zero or negligible dual variables can be safely removed without significantly weakening the cut. This decision requires a careful heuristic that considers not only the current values of the dual variables but also the overall structure of the problem and the potential future impact of these neurons. By intelligently selecting which $z$ variables to exclude, we aim to produce a stronger cut that is both effective in pruning the search space and efficient in computing. To do this, we use a heuristic to determine whether each neuron should be tentatively dropped from the list of constraints. Algorithm 1 shows the constraint strengthening with the neuron elimination heuristic.

**Neuron Elimination Heuristic for Constraint Strengthening** First, we compute a heuristic influence score for each neuron to assess its impact on the verification objective. This score is based on the improvement in the lower bound of $f^\star$ before and after introducing the neuron's split in the BaB tree. By recording the computed lower bounds at each node, we can measure how beneficial each constraint is to the verification process. Second, we rank the neurons according to their influence scores and tentatively drop those that contribute least to improving the lower bound. We will retain neurons whose corresponding Lagrange multipliers, $\mu$ or $\tau$, are greater than zero. For neurons with $\mu$ or $\tau$ equal to zero, we remove a certain percentile of neurons with the lowest scores, as indicated by the "drop_percentage" parameter in Algorithm 1. When a neuron is dropped, its split is canceled. Then, we perform a re-verification step using only the reduced subset of constraints. This involves recomputing the lower bound on $f^\star$ based solely on the selected constraints. If the verification still succeeds—that is, the lower bound remains non-negative—the reduced set induces a new cutting plane. This new cutting plane is applied to all subproblems within the BaB process after further strengthening. Finally, we can iteratively repeat the process of reducing the constraint set, aiming to generate even stronger cuts. The process terminates when the property can no longer be verified with the current subset of constraints. This iterative refinement is designed to focus on the most influential neurons, potentially enhancing the efficiency of the verification. Once the new cuts determined, we merge pairs of cuts if possible (e.g. merging $z_1 + z_2 \geq 0$ and $z_1 - z_2 \geq 0$ to $z_1 \geq 0$).

---

**Algorithm 1** Constraint Strengthening

---

**Require:** $f$: model; $\underline{f}$: lower bound; $\mathcal{D}$: Domain
  $\hat{\mathcal{Z}} \in \mathcal{D}$ : UNSAT split constraints set;   $\mathcal{T}, \mathcal{M} \in \mathcal{D}$ : $\boldsymbol{\tau}$ set for $\hat{\mathcal{Z}}_-$ and $\boldsymbol{\mu}$ set for $\hat{\mathcal{Z}}_+$
  $\mathcal{C}_{\text{cut}}$ : current set of known cuts;   drop_percentage : Percentage of splits to be dropped
1: neuron_influence_scores $\leftarrow$ Neuron_Elinimation_Heuristic($\underline{f}, \mathcal{D}$)
2: score_threshold $\leftarrow$ Percentile(neuron_influence_scores, drop_percentage)
3: $\mathcal{Z}_{\text{new}} = \emptyset$
4: **for** $i \in [|\text{neuron\_influence\_scores}|]$ **do**
5:   **if** $\hat{\mathcal{Z}}_i \in \mathcal{T} \cup \mathcal{M} \neq 0$ or neuron_influence_scores$_i \geq$ score_threshold **then**
6:     $\mathcal{Z}_{\text{new}} \leftarrow \mathcal{Z}_{\text{new}} \cup \hat{\mathcal{Z}}_i$
7:     $\underline{f}_{\text{re-verification}} \leftarrow$ Solve_Bound($f, \mathcal{C}_{\text{cut}} \cup \mathcal{Z}_{\text{new}}$)
8: **if** $\underline{f}_{\text{re-verification}} \geq 0$ **then**
9:   strengthened_cut $\leftarrow$ Infer_Cut($\mathcal{Z}_{\text{new}}$)
10:   $\mathcal{C}_{\text{cut}} \leftarrow \mathcal{C}_{\text{cut}} \cup \{\text{strengthened\_cut}\}$
11:   $\mathcal{C}_{\text{cut}} \leftarrow$ Constraint_Strengthening($f, \underline{f}, \mathcal{D}, \mathcal{Z}_{\text{new}}, \mathcal{C}_{\text{cut}}, \text{drop\_percentage}$)
12: $\mathcal{C}_{\text{cut}} \leftarrow$ Merge_Cuts($\mathcal{C}_{\text{cut}}$)
13: **return** $\mathcal{C}_{\text{cut}}$

---

**Multi-Tree Search** Traditionally, the BaB process generates a single search tree. We propose augmenting this approach by performing multiple BaB processes in parallel as a presolving step, with each process exploring a different set of branching decisions. At each branching point, we initialize multiple trees and apply various branching decisions simultaneously. While this initially increases the number of subproblems, the cutting planes generated in one tree are universally valid and can be applied to all other trees. These newly introduced cuts can help prove UNSAT for nodes in other trees, thereby inducing additional cutting planes and amplifying the pruning effect across the entire search space, illustrated in Fig. 2c.

Since computational resources must be allocated across multiple trees, we prioritize nodes for further expansion that have the highest lower bound on the optimization objective. This strategy ensures that more promising trees receive more computational resources. After a predefined short timeout, we

consolidate our efforts by pruning all but one of the trees and proceed with the standard BaB process augmented with BICCOS on the selected tree. We choose the tree that has been expanded the most frequently, as this indicates that its bounds are closest to verifying the property.

**BaB Tree Searching Strategy** In the standard BaB process (e.g., in $\beta$-CROWN), branches operate independently without sharing information, so the order in which they are explored does not affect the overall runtime. For memory access efficiency, there is a slight preference for implementing BaB as a depth-first search (DFS), where constraints are added until unsatisfiability (UNSAT) can be proven [61, 54]. This approach focuses the search on deeper branches before returning to shallower nodes.

However, in the context of BICCOS, our objective is to generate strong cutting planes that can prune numerous branches across different subproblems. To maximize the generality of these cutting planes, they need to be derived from UNSAT nodes with as few constraints as possible. While constraint strengthening techniques can simplify the constraints, this process is more straightforward when the original UNSAT node already has a minimal set of constraints. Even if only a few constraints are eliminated, the resulting cutting plane can significantly impact many other subproblems. To facilitate this, we propose performing the BaB algorithm using a breadth-first search (BFS) strategy. By exploring nodes that are closest to the root and have the fewest neuron constraints, we can generate more general and impactful cutting planes earlier in the search process.

---

**Algorithm 2** Branch-and-bound Inferred Cuts with Constraint Strengthening (BICCOS).

---

**Require:** $f$: model; $n$: batch size; time out threshold
1: $\mathcal{D}_{\text{Unknown}}, \underline{f} \leftarrow \text{Init}(f, \emptyset)$
2: $\mathcal{D}_{\text{Unknown}}, \mathcal{C}_{\text{inferred\_cuts}} \leftarrow \text{Multi\_Tree\_Search}(f, \emptyset)$
3: **while** $|\mathcal{D}_{\text{Unknown}}| > 0$ and not timed out **do**
4: $\quad (\mathcal{Z}_1, \ldots, \mathcal{Z}_n) \leftarrow \text{Batch\_Pick\_Out}_{BFS}(\mathcal{D}_{\text{Unknown}}, n)$
5: $\quad (\mathcal{Z}_1^-, \mathcal{Z}_1^+, \ldots, \mathcal{Z}_n^-, \mathcal{Z}_n^+) \leftarrow \text{Batch\_Split}(\mathcal{Z}_1, \ldots, \mathcal{Z}_n)$
6: $\quad (\underline{f}_{\mathcal{Z}_1^-}, \underline{f}_{\mathcal{Z}_1^+}, \ldots, \underline{f}_{\mathcal{Z}_n^-}, \underline{f}_{\mathcal{Z}_n^+}) \leftarrow \text{Solve\_Bound}(f, \mathcal{C}_{\text{inferred\_cuts}}, \mathcal{Z}_1^-, \mathcal{Z}_1^+, \ldots, \mathcal{Z}_n^-, \mathcal{Z}_n^+)$
7: $\quad \mathcal{D}_{\text{UNSAT}} \leftarrow \text{Domain\_Filter}_{\text{UNSAT}}([\underline{f}_{\mathcal{Z}_1^-}, \mathcal{Z}_1^-], [\underline{f}_{\mathcal{Z}_1^+}, \mathcal{Z}_1^+], \ldots, [\underline{f}_{\mathcal{Z}_n^+}, \mathcal{Z}_n^+])$
8: $\quad$ **for all** $\mathcal{D}_i \in \mathcal{D}_{\text{UNSAT}}$ **do**
9: $\quad\quad \mathcal{C}_{\text{inferred\_cuts}} \leftarrow \text{Constraint\_Strengthening}(f, \underline{f}_i, \mathcal{D}_i, \text{drop\_percentage})$
10: $\quad \mathcal{D}_{\text{Unknown}} \leftarrow \mathcal{D}_{\text{Unknown}} \bigcup \text{Domain\_Filter}_{\text{Unknown}}([\underline{f}_{\mathcal{Z}_1^-}, \mathcal{Z}_1^-], [\underline{f}_{\mathcal{Z}_1^+}, \mathcal{Z}_1^+], \ldots, [\underline{f}_{\mathcal{Z}_n^+}, \mathcal{Z}_n^+])$
11: **return** UNSAT if $|\mathcal{D}_{\text{Unknown}}| = 0$ else Unknown

---

### 3.3 BICCOS Summary

Algorithm 2 summarizes our proposed BICCOS algorithm, with the modifications to the standard BaB algorithm highlighted in brown. First (line 2), instead of exploring a single tree, we explore multiple trees in parallel as a presolving step. This process may involve constraint strengthening and utilizes cut inference analogous to the procedures in lines 3-15 of the algorithm. After several iterations, we prune all but one of the trees. From this point forward, only the selected tree is expanded further, following the regular BaB approach.

Until all subdomains of this tree have been verified, BICCOS selects batches of unverified subdomains add additional branching decisions, and attempt to prove the verification property. Unlike regular BaB, it then applies constraint strengthening to all identified UNSAT nodes and infers the corresponding cutting planes. These cutting planes are added to all currently unverified subdomains, potentially improving their lower bounds enough to complete the verification process. If BICCOS fails to identify helpful cutting planes, it effectively behaves like the regular BaB algorithm. We have implemented BICCOS in the $\alpha,\beta$-CROWN toolbox. Notably, the cuts found by BICCOS are compatible with those from MIP solvers in GCP-CROWN, and all cuts can be combined in cases where MIP cuts are beneficial.

## 4 Experiments

We evaluate our verifier, BICCOS, on several popular verification benchmarks from VNN-COMP [4, 40, 10] and on the SDP-FO benchmarks used in multiple studies [15, 54, 41]. In the following discussion, $\alpha,\beta$-CROWN refers to the verification tool that implements various verification techniques, while $\beta$-CROWN and GCP-CROWN denote specific algorithms implemented within $\alpha,\beta$-CROWN. To ensure the comparability of our method's effects and to minimize the influence of hardware and equipment advances, we rerun $\beta$-CROWN and GCP-CROWN with MIP cuts for each experiment. Additionally, we use the same BaB algorithm as in $\beta$-CROWN and employ filtered smart branching

Table 1: Comparison of different toolkits and BICCOS on VNN-COMP benchmarks. Results on non-CROWN or BICCOS were run on different hardware. "-" indicates that the benchmark was not supported.

| Method | oval22 time(s) | oval22 # verified | cifar100-tinyimagenet-2022 time(s) | cifar100-tinyimagenet-2022 # verified | cifar100-2024 time(s) | cifar100-2024 # verified | tinyimagenet-2024 time(s) | tinyimagenet-2024 # verified |
|---|---|---|---|---|---|---|---|---|
| nnenum* [3, 5] | 630.06 | 3 | - | - | - | - | - | - |
| Marabou†‡ [32, 56] | 429.13 | 5 | 186.11 | 27 | - | 0 | - | 0 |
| ERAN*[41, 39] | 233.84 | 6 | - | - | - | - | - | - |
| OVAL* [17, 16] | 393.14 | 11 | - | - | - | - | - | - |
| Venus2 [9, 34] | 386.71 | 17 | - | - | - | - | - | - |
| VeriNet† [28, 29] | 73.65 | 17 | 39.43 | 69 | - | - | - | - |
| MN-BaB† [21] | 137.13 | 19 | 40.27 | 36 | - | - | - | - |
| PyRAT‡ [25] | - | - | - | - | 42.38 | 68 | 55.64 | 49 |
| $\beta$-CROWN [62, 59, 54] | 23.26 | 20 | 11.95 | 69 | 15.48 | 119 | 28.87 | 135 |
| GCP-CROWN with MIP cuts [61] | 32.12 | 25 | 18.42 | 69 | 19.32 | 119 | 31.60 | 134 |
| BICCOS | 59.84 | **26** | 13.38 | **72** | 13.58 | **125** | 16.33 | **140** |
| Upper Bound | | 27 | | 94 | | 168 | | 157 |

* Results from VNN-COMP 2021 report [4].    † Results from VNN-COMP 2022 report [43]    ‡ Results from VNN-COMP 2024 website[1]

Table 2: Verified accuracy (Ver.%) and avg. per-example verification time (s) on 7 models from [15].

| Dataset | Model $\epsilon = 0.3$ and $\epsilon = 2/255$ | PRIMA [41] Ver.% | PRIMA [41] Time | $\beta$-CROWN [54] Ver.% | $\beta$-CROWN [54] Time(s) | MN-BaB [21]* Ver.% | MN-BaB [21]* Time(s) | Venus2 [9, 34] Ver.% | Venus2 [9, 34] Time(s) | GCP-CROWN (MIP cuts) [61] Ver.% | GCP-CROWN (MIP cuts) [61] Time(s) | BICCOS Ver.% | BICCOS Time(s) | Upper bound |
|---|---|---|---|---|---|---|---|---|---|---|---|---|---|---|
| MNIST | CNN-A-Adv | 44.5 | 135.9 | 71.0 | 3.53 | - | - | 35.5 | 148.4 | 70.5 | 7.34 | **75.5** | 13.37 | 76.5 |
| CIFAR | CNN-A-Adv | 41.5 | 4.8 | 45.5 | 5.17 | 42.5 | 68.3 | 47.5 | 26.0 | 48.5 | 4.78 | **49.0** | 8.94 | 50.0 |
| | CNN-A-Adv-4 | 45.0 | 4.9 | 46.5 | 0.78 | 46.0 | 37.7 | 47.5 | 13.1 | 48.0 | 1.47 | **48.5** | 1.81 | 49.5 |
| | CNN-A-Mix | 37.5 | 34.3 | 42.5 | 4.78 | 35.0 | 140.3 | 33.5 | 72.4 | 47.5 | 9.70 | **48.5** | 10.34 | 53.0 |
| | CNN-A-Mix-4 | 48.5 | 7.0 | 51.0 | 0.79 | 49.0 | 70.9 | 49.0 | 37.3 | 54.5 | 3.82 | **56.0** | 5.23 | 57.5 |
| | CNN-B-Adv | 38.0 | 343.6 | 47.5 | 6.39 | - | - | - | - | 49.0 | 10.07 | **54.5** | 17.75 | 65.0 |
| | CNN-B-Adv-4 | 53.5 | 43.8 | 56.0 | 3.20 | - | - | - | - | 58.5 | 9.63 | **62.0** | 8.27 | 63.5 |

* MN-BaB with 600s timeout threshold for all models. "-" indicates that we could not run a model due to unsupported model structure or other errors. We run $\beta$-CROWN, GCP-CROWN with MIP cuts and BICCOS with a shorter 200s timeout for all models. The increased timeout for MN-BaB may increase the percentage of verified instances. However, we can still achieve better verified accuracy than all other baselines. Other results are reported from [61].

(FSB) [16] as the branching heuristic in all experiments. Note that in experiments GCP-CROWN refers to GCP-CROWN solver with MIP cuts. We also conduct ablation studies to identify which components of BICCOS contribute the most, including analyses of verification accuracy & time, number of cuts generated, and number of domains visited. Experimental settings are described in Appendix C.1.

**Results on VNN-COMP benchmarks**    We first evaluate BICCOS on many challenging benchmarks with large models, including two VNN-COMP 2024 benchmarks: `cifar100-2024` and `tinyimagenet-2024`; two VNN-COMP 2022 benchmarks: `cifar100-tinyimagenet-2022` and `oval22`. Shown in Table 1, our proposed method, BICCOS, outperforms most other verifiers on the tested benchmarks, achieving the highest number of verified instances in four benchmark sets. BICCOS consistently outperforms the baseline $\alpha, \beta$-CROWN verifier (the $\beta$-CROWN and GCP-CROWN (MIP cuts) lines), verifying more instances across almost all benchmark sets. In particular, GCP-CROWN with MIP cuts cannot scale to the larger network architectures in the `cifar100` and `tinyimagenet` benchmarks with network sizes between 14.4 and 31.6 million parameters, due to its reliance on an MIP solver. BICCOS, on the other hand, can infer cutting planes without the need for an MIP solver and noticeably outperforms the baseline on `cifar100` and `tinyimagenet`. Note that the increase in average runtime (e.g., on the `cifar100-tinyimagenet-2022` benchmark) is expected. The instances that could not be verified at all previously but can be verified using BICCOS tend to require runtimes that are below the timeout but above the baseline's average runtime.

**Results on SDP-FO benchmarks**    We further evaluated BICCOS, on the challenging SDP-FO benchmarks introduced in previous studies [15, 54]. These benchmarks consist of seven predominantly adversarial trained MNIST and CIFAR models, each containing numerous instances that are difficult for many existing verifiers. Our results, detailed in Table 2, demonstrate that BICCOS significantly improves verified accuracy across all the tested models when compared to current state-of-the-art verifiers. On both MNIST and CIFAR dataset, BICCOS not only surpasses the performance of methods like $\beta$-CROWN and GCP-CROWN on the CNN-A-Adv model but also approaches the empirical robust accuracy upper bound, leaving only a marginal gap. A slight increase in average time in some cases is attributed to the higher number of solved instances.

**Ablation Studies on BICCOS Components.**    To evaluate the contributions of individual components of BICCOS, we performed ablation studies summarized in Table 3. The BICCOS base version with BaB inferred cuts and constraint strengthening already shows competitive performance in many models. Cuts from MIP solvers are compatible with BICCOS and can be added for smaller models that can be handled by MIP solver. Integrating Multi-Tree Search (MTS) significantly boosts performance. On the CIFAR CNN-B-Adv model, verified accuracy rises to 54.5%, outperforming

Table 3: Ablation Studies on Verified accuracy (Var.%), avg. per-example verification time (s) analysis for all method verified instances on different BICCOS components.

| Dataset $\epsilon = 0.3$ and $\epsilon = 2/255$ | Model | $\beta$-CROWN [54] Ver.% | Time (s) | GCP-CROWN(MIP cuts) [61] Ver.% | Time(s) | BICCOS (base) Ver.% | Time(s) | BICCOS (with MTS) Ver.% | Time(s) | BICCOS (auto) Ver.% | Time(s) | Upper bound |
|---|---|---|---|---|---|---|---|---|---|---|---|---|
| MNIST | CNN-A-Adv | 71.0 | 3.53 | 70.5 | 7.34 | **76.5** | 5.61 | **76.5** | 8.86 | 75.5 | 13.37 | 76.5 |
| CIFAR | CNN-A-Adv | 45.5 | 5.17 | 48.5 | 4.78 | 47.5 | 4.88 | 47.5 | 5.01 | **49.0** | 4.26 | 50.0 |
| | CNN-A-Adv-4 | 46.5 | 0.78 | 48.0 | 1.47 | 48.0 | 1.27 | 47.5 | 1.15 | **48.5** | 1.81 | 49.5 |
| | CNN-A-Mix | 42.5 | 4.78 | 47.5 | 9.70 | 47.0 | 6.68 | 47.0 | 7.87 | **48.5** | 10.31 | 53.0 |
| | CNN-A-Mix-4 | 51.0 | 0.79 | 54.5 | 3.82 | 55.0 | 6.96 | 54.0 | 2.87 | **56.0** | 5.23 | 57.5 |
| | CNN-B-Adv | 47.5 | 6.39 | 49.0 | 10.07 | 52.0 | 8.14 | 52.5 | 10.13 | **54.5** | 17.75 | 65.0 |
| | CNN-B-Adv-4 | 56.0 | 3.20 | 58.5 | 8.27 | 60.0 | 3.18 | 60.5 | 4.38 | **62.0** | 9.63 | 63.5 |
| oval22 | | 66.67 | 23.26 | 83.33 | 32.12 | 73.33 | 18.75 | 70.00 | 17.23 | **86.66** | 59.84 | 90.0 |
| cifar100-2024 | | 59.5 | 15.48 | 59.5 | 19.32 | **62.5** | 12.74 | 61.5 | 12.18 | **62.5** | 13.57 | 84.0 |
| tinyimagenet-2024 | | 67.5 | 28.87 | 67.0 | 31.60 | **70.0** | 13.84 | **70.0** | 17.8 | **70.0** | 16.32 | 78.5 |

* We run our BICCOS in different ablation studies with a shorter 200s timeout for all models and compare it to $\beta$-CROWN and GCP-CROWN, it achieves better verified accuracy than all other baselines.

GCP-CROWN with MIP cuts's 49%. We also design an adaptive BICCOS configuration (BICCOS auto), which automatically turns on MTS and/or MIP-based cuts according to neural network and verification problem size and quantity, achieves the highest verified accuracies across most of models and is used as the default option of the verifier when BICCOS is enabled. A detailed table with the numbers of cuts and domains visited is provided in Appendix C.2.

## 5 Related work

Our work is based on the branch and bound framework for neural network verification [12, 17, 59, 28, 54, 16, 37, 21], which is one of the most popular approaches that lead to state-of-the-art results [4, 10, 40]. Most BaB-based approaches do not consider the correlations among subproblems - for example, in $\beta$-CROWN [54], the order of visiting the nodes in the BaB search tree does not change the verification outcome as the number of leaf nodes will be the same regardless of how the leaves are split. Our work utilizes information on the search tree and can gather more effective cuts when shallower nodes are visited first.

Exploring the dependency or correlations among neurons has also been identified as a potential avenue to enhance verification bounds. While several studies have investigated this aspect [2, 16, 42, 50], their focus has primarily been on improving the bounding step without explicitly utilizing the relationships among ReLUs during the branching process. Venus [9] considers the implications among neurons with constraints similar to our cutting planes. However, their constraints were not discovered using the verified subproblems during BaB or multi-tree search, and cannot be strengthened. On the other hand, cutting plane methods encode dependency among neurons as general constraints [61, 35], and our work developed a new cutting plane that can be efficiently constructed and strengthened during BaB, the first time in literature.

In addition, some NN verifiers are based on the satisfiability modulo theories (SMT) formulations [32, 46, 19, 36], which may internally use an SAT-solving procedure [8] such as DPLL [22] or CDCL [38]. These procedures may discover conflicts in boolean variable assignments, corresponding to eliminating certain subproblems in BaB. However, they differ from BICCOS in two significant aspects: first, although DPLL or CDCL may discover constraints to prevent some branches with neurons involved in these constraints, they cannot efficiently use these constraints as cutting planes that may tighten the bounds for subproblems never involving these neurons; second, DPLL or CDCL works on the abstract problem where each ReLU is represented as a boolean variable, and cannot take full advantage of the underlying bound propagation solver to strengthen constraints as we did in Alg. 1. Based on our observation in Sec. 3, the constraints discovered during BaB are often unhelpful without strengthening unless in a different search tree, so their effectiveness is limited. However, the learned conflicts can be naturally translated into cuts 3, making this a future work.

More related works on SMT, MIP solvers, nogood learning and cutting plane method in VNN [18, 45, 26, 19, 11, 2, 47, 52, 30, 61, 35] are discussed in Appendix D.

## 6 Conclusion

We exploit the structure of the NN verification problem to generate efficient and scalable cutting planes, leveraging neuron relationships within verified subproblems in a branch-and-bound search tree. Our experimental results demonstrate that the proposed BICCOS algorithm achieves very good scalability while outperforming many other tools in the VNN-COMP, and can solve benchmarks that existing methods utilizing cutting planes could not scale to. Limitations are discussed in Appendix E.

**Acknowledgment** Huan Zhang is supported in part by the AI2050 program at Schmidt Sciences (AI2050 Early Career Fellowship) and NSF (IIS-2331967). Grani A. Hanasusanto is supported in part by NSF (CCF-2343869 and ECCS-2404413). Computations were performed with computing resources granted by RWTH Aachen University under project rwth1665. We thank the anonymous reviewers for helping us improve this work.

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

# Appendix

In Sec. A, we introduce the Mixed Integer Programming (MIP) formulation for Neural Network Verification and explain how to get the lower bound by solving the LP relaxation and Planet relaxation. In Sec. B, we provide the complete proof for Proposition 3.1 and Theorem 3.2 and Corollary 3.3. Then, in Sec. C, we present the configuration, ablation study, and additional experimental results. Finally in Sec. D and Sec. E, we discuss more related works and limitations of our method.

## A  MIP Formulation and LP & Planet Relaxation

**The MIP Formulation**   The mixed integer programming (MIP) formulation is the root of many NN verification algorithms. Given the ReLU activation function's piecewise linearity, the model requires binary encoding variables, or ReLU indicators $z$ only for unstable neurons. We formulate the optimization problem aiming to minimize the function $f(x)$, subject to a set of constraints that encapsulate the DNN's architecture and the perturbation limits around a given input $x$, as follows:

$$f^\star = \min_{x, \hat{x}, z} f(x) \quad \text{s.t. } f(x) = x^{(L)}; \hat{x}^{(0)} = x \in \mathcal{X}; \tag{4}$$

$$x^{(i)} = \mathbf{W}^{(i)}\hat{x}^{(i-1)} + \mathbf{b}^{(i)}; \quad i \in [L], \tag{5}$$

$$\mathcal{I}^{+(i)} := \{j : l_j^{(i)} \geq 0\}, \mathcal{I}^{-(i)} := \{j : u_j^{(i)} \leq 0\}, \mathcal{I}^{(i)} := \{j : l_j^{(i)} \leq 0, u_j^{(i)} \geq 0\}, \tag{6}$$

$$\mathcal{I}^{+(i)} \cup \mathcal{I}^{-(i)} \cup \mathcal{I}^{(i)} = \mathcal{J}^{(i)} \tag{7}$$

$$\hat{x}_j^{(i)} \geq 0; j \in \mathcal{I}^{(i)}, i \in [L-1] \tag{8}$$

$$\hat{x}_j^{(i)} \geq x_j^{(i)}; j \in \mathcal{I}^{(i)}, i \in [L-1] \tag{9}$$

$$\hat{x}_j^{(i)} \leq u_j^{(i)} z_j^{(i)}; j \in \mathcal{I}^{(i)}, i \in [L-1] \tag{10}$$

$$\hat{x}_j^{(i)} \leq x_j^{(i)} - l_j^{(i)}(1 - z_j^{(i)}); j \in \mathcal{I}^{(i)}, i \in [L-1] \tag{11}$$

$$z_j^{(i)} \in \{0, 1\}; j \in \mathcal{I}^{(i)}, i \in [L-1] \tag{12}$$

$$\hat{x}_j^{(i)} = x_j^{(i)}; j \in \mathcal{I}^{+(i)}, i \in [L-1] \tag{13}$$

$$\hat{x}_j^{(i)} = 0; j \in \mathcal{I}^{-(i)}, i \in [L-1]. \tag{14}$$

Here, the set $\mathcal{J}^{(i)}$ comprises all neurons in the layer $i$, which are further categorized into three distinct classes: 'active' ($\mathcal{I}^{+(i)}$), 'inactive' ($\mathcal{I}^{-(i)}$), and 'unstable' ($\mathcal{I}^{(i)}$); we further let $l_j^{(i)} = \min_{x \in \mathcal{X}} f_j^{(i)}(x), u_j^{(i)} = \max_{x \in \mathcal{X}} f_j^{(i)}(x), \quad \forall j \in \mathcal{J}^{(i)}, i \in [L-1]$. The MIP approach is initialized with pre-activation bounds $l^{(i)} \leq x^{(i)} \leq u^{(i)}$ for each neuron within the feasible set $\mathcal{X}$, across all layers $i$. These bounds can be calculated recursively through the MIP from the first layer to the final layer. However, MIP problems are generally NP-hard as they involve integer variables.

**The LP and Planet relaxation**   By relaxing the binary variables 12 to $z_j^{(i)} \in [0, 1], j \in \mathcal{I}^{(i)}, i \in [L-1]$, we can get the LP relaxation formulation. By replacing the constraints in 10, 11, 12 with

$$\hat{x}_j^{(i)} \leq \frac{u_j^{(i)}}{u_j^{(i)} - l_j^{(i)}}(x_j^{(i)} - l_j^{(i)}); \quad j \in \mathcal{I}^{(i)}, i \in [L-1] \tag{15}$$

we can eliminate the $z$ variables and get the well-known Planet relaxation formulation. Both of these two relaxations are solvable in polynomial time to yield lower bounds.

**Lemma A.1.** *In the LP relaxation, splitting an unstable ReLU neuron $(i, j)$ based on $z_j^{(i)} = 0$ is equivalent to setting $x_j^{(i)} \leq u_j^{(i)} = 0$ in the Planet relaxation with the relaxed variable $z \in [0, 1]$ projected out. Correspondingly, splitting based on $z_j^{(i)} = 1$ is equivalent to setting $x_j^{(i)} \geq l_j^{(i)} = 0$.*

*Proof:*   An unstable neuron $(i, j)$ is one where the pre-activation bounds satisfy $l_j^{(i)} < 0 < u_j^{(i)}$. It could be represented as the following system in LP relaxation: 8, 10, 11, 12, and the system in Planet

relaxation: 8, 15. We will now analyze the two cases of fixing $z_j^{(i)}$ to 0 and 1 in the LP relaxation and show their equivalence to modifying the bounds $u_j^{(i)}$ and $l_j^{(i)}$ in the Planet relaxation.

1. $z_j^{(i)} = 0$ in LP relaxation $\Leftrightarrow u_j^{(i)} = 0$ in Planet Relaxation.

   ($\Rightarrow$) Assume $z_j^{(i)} = 0$ in LP relaxation. From 10, $\hat{x}_j^{(i)} \leq u_j^{(i)} \times 0 = 0 \implies \hat{x}_j^{(i)} \leq 0$. From 8: $\hat{x}_j^{(i)} \geq 0$. Combining the above, we get $\hat{x}_j^{(i)} = 0$. From 11, $\hat{x}_j^{(i)} \leq x_j^{(i)} - l_j^{(i)}(1 - 0) = x_j^{(i)} - l_j^{(i)}$. Since $\hat{x}_j^{(i)} = 0$ and 9, this simplifies to $0 \leq x_j^{(i)} - l_j^{(i)} \implies 0 \geq x_j^{(i)} \geq l_j^{(i)}$ This is consistent with the pre-activation bounds of the neuron. Therefore, fixing $z_j^{(i)} = 0$ in the LP relaxation forces the activation $\hat{x}_j^{(i)}$ to be zero, effectively modeling the neuron as inactive.

   ($\Leftarrow$) Assume $u_j^{(i)} = 0$ in the Planet relaxation. Set $u_j^{(i)} = 0$. The constraint 15 becomes $\hat{x}_j^{(i)} \leq \frac{0}{0 - l_j^{(i)}}(x_j^{(i)} - l_j^{(i)}) = 0$. From 8 $\hat{x}_j^{(i)} \geq 0$. Combining the above, we get $\hat{x}_j^{(i)} = 0$. Thus, setting $u_j^{(i)} = 0$ in the Planet relaxation also forces $\hat{x}_j^{(i)}$ to zero, mirroring the effect of fixing $z_j^{(i)} = 0$ in the LP relaxation.

2. $z_j^{(i)} = 1 \Leftrightarrow l_j^{(i)} = 0$. The proof follows a similar step. $\qquad\qquad\square$

# B  Proof of Main Results

## B.1  Proof of Proposition 3.1

*Proof:*  Due to the relationship between the $x$ and $z$ in Lemma A.1, i.e. for any unstable neuron $i$, $x_i \geq l_i = 0 \Longleftrightarrow z_i = 1$ and $x_i \leq u_i = 0 \Longleftrightarrow z_i = 0 \Longleftrightarrow (1 - z_i) = 1$, we have the following equation to represent an UNSAT path:

$$\sum_{i \in \mathcal{Z}^+} z_i + \sum_{i \in \mathcal{Z}^-}(1 - z_i) = |\mathcal{Z}^+| + |\mathcal{Z}^-| \tag{16}$$

By taking negation of Eq 16 to exclude this path, we have:

$$\sum_{i \in \mathcal{Z}^+} z_i + \sum_{i \in \mathcal{Z}^-}(1 - z_i) > |\mathcal{Z}^+| + |\mathcal{Z}^-| \tag{17}$$

$$\sum_{i \in \mathcal{Z}^+} z_i + \sum_{i \in \mathcal{Z}^-}(1 - z_i) < |\mathcal{Z}^+| + |\mathcal{Z}^-| \tag{18}$$

Notice that $z \in [0, 1]^{|\mathcal{Z}|}$, implies that the sum of $z$ should be bounded by its cardinality, $\sum_{i \in \mathcal{Z}^+} z_i + \sum_{i \in \mathcal{Z}^-}(1 - z_i) \leq |\mathcal{Z}^+| + |\mathcal{Z}^-|$, and the binary indicator property, thus we only consider inequality 18:

$$\sum_{i \in \mathcal{Z}^+} z_i + \sum_{i \in \mathcal{Z}^-}(1 - z_i) = \sum_{i \in \mathcal{Z}^+} z_i + |\mathcal{Z}^-| - \sum_{i \in \mathcal{Z}^-} z_i \leq |\mathcal{Z}^+| + |\mathcal{Z}^-| - 1 \tag{19}$$

Thus, the BaB inferred cut 3 can be used to exclude this UNSAT status. $\qquad\qquad\square$

## B.2  Proof of Theorem 3.2

To derive the bound propagation, first, we assign Lagrange dual variables to each non-trivial constraint in LP relaxation with any BaB split set $\mathcal{Z}$. Next, we analyze how splitting on the ReLU indicator $z$ influences the generation of these dual variables. This analysis helps us determine the impact of the splits on the dual formulation. Then, we formulate the Lagrangian dual as a min-max problem. By applying strong duality and eliminating the inner minimization problem, we can simplify the dual formulation. Finally, this results in a dual formulation that we use to perform bound propagation effectively.

**Theorem 3.2.** Given any BaB split set $\mathcal{Z}$, optimizable parameters $0 \leq \alpha_j^{(i)} \leq 1$ and $\boldsymbol{\beta}, \boldsymbol{\mu}, \boldsymbol{\tau} \geq 0$, ${\pi_j^{(i)}}^{\star}$ is a function of $\boldsymbol{Q}_{:,j}^{(i)}$:

$$g(\boldsymbol{\alpha}, \boldsymbol{\beta}, \boldsymbol{\mu}, \boldsymbol{\tau}) = -\epsilon \|\boldsymbol{\nu}^{(1)\top} \mathbf{W}^{(1)} \boldsymbol{x}_0\|_1 - \sum_{i=1}^{L} \boldsymbol{\nu}^{(i)\top} \mathbf{b}^{(i)} - \boldsymbol{\beta}^\top \boldsymbol{d} + \sum_{i=1}^{L-1} \sum_{j \in \mathcal{I}^{(i)}} h_j^{(i)}(\boldsymbol{\beta})$$

where variables $\boldsymbol{\nu}^{(i)}$ are obtained by propagating $\boldsymbol{\nu}^{(L)} = -1$ throughout all $i \in [L-1]$:

$$\nu_j^{(i)} = \boldsymbol{\nu}^{(i+1)\top} \mathbf{W}_{:,j}^{(i+1)} - \boldsymbol{\beta}^\top (\boldsymbol{H}_{:,j}^{(i)} + \boldsymbol{G}_{:,j}^{(i)}), \; j \in \mathcal{I}^{+(i)}$$

$$\nu_j^{(i)} = \boldsymbol{\nu}^{(i+1)\top} \mathbf{W}_{:,j}^{(i+1)} - \boldsymbol{\beta}^\top (\boldsymbol{H}_{:,j}^{(i)} + \boldsymbol{G}_{:,j}^{(i)}) + \mu_j^{(i)}, \; j \in \mathcal{Z}^{+(i)},$$

$$\nu_j^{(i)} = -\boldsymbol{\beta}^\top \boldsymbol{H}_{:,j}^{(i)}, \; j \in \mathcal{I}^{-(i)},$$

$$\nu_j^{(i)} = -\boldsymbol{\beta}^\top \boldsymbol{H}_{:,j}^{(i)} - \tau_j^{(i)}, \; j \in \mathcal{Z}^{-(i)},$$

$$\nu_j^{(i)} = {\pi_j^{(i)}}^{\star} + \alpha_j^{(i)} [\hat{\nu}_j^{(i)}]_- - \boldsymbol{\beta}^\top \boldsymbol{H}_{:,j}^{(i)}, \; j \in \mathcal{I}^{(i)} \setminus \mathcal{Z}^{(i)}$$

Here $\hat{\nu}_j^{(i)}$, ${\pi_j^{(i)}}^{\star}$ and $\tau_j^{(i)}$ are defined for each unstable and without split neuron $j \in \mathcal{I}^{(i)} \setminus \mathcal{Z}^{(i)}$, and $h_j^{(i)}(\boldsymbol{\beta})$ is defined for all unstable neurons $j \in \mathcal{I}^{(i)}$.

$$[\hat{\nu}_j^{(i)}]_+ := \max(\boldsymbol{\nu}^{(i+1)\top} \mathbf{W}_{:,j}^{(i+1)} - \boldsymbol{\beta}^\top \boldsymbol{G}_{:,j}^{(i)}, 0), \; [\hat{\nu}_j^{(i)}]_- := \min(\boldsymbol{\nu}^{(i+1)\top} \mathbf{W}_{:,j}^{(i+1)} - \boldsymbol{\beta}^\top \boldsymbol{G}_{:,j}^{(i)}, 0)$$

$${\pi_j^{(i)}}^{\star} = \max\left(\min\left(\frac{u_j^{(i)} [\hat{\nu}_j^{(i)}]_+ - \boldsymbol{\beta}^\top \boldsymbol{Q}_{:,j}^{(i)}}{u_j^{(i)} - l_j^{(i)}}, [\hat{\nu}_j^{(i)}]_+\right), 0\right), j \in \mathcal{I}^{(i)} \setminus \mathcal{Z}^{(i)}$$

$$h_j^{(i)}(\boldsymbol{\beta}) = \begin{cases} l_j^{(i)} {\pi_j^{(i)}}^{\star} & \text{if } l_j^{(i)} [\hat{\nu}_j^{(i)}]_+ \leq \boldsymbol{\beta}^\top \boldsymbol{Q}_{:,j}^{(i)} \leq u_j^{(i)} [\hat{\nu}_j^{(i)}]_+ \text{ and } j \in \mathcal{I}^{(i)} \setminus \mathcal{Z}^{(i)}, \\ 0 & \text{if } \boldsymbol{\beta}^\top \boldsymbol{Q}_{:,j}^{(i)} \geq u_j^{(i)} [\hat{\nu}_j^{(i)}]_+ \text{ or } j \in \mathcal{Z}^{-(i)} \\ \boldsymbol{\beta}^\top \boldsymbol{Q}_{:,j}^{(i)} & \text{if } \boldsymbol{\beta}^\top \boldsymbol{Q}_{:,j}^{(i)} \leq l_j^{(i)} [\hat{\nu}_j^{(i)}]_+ \text{ or } j \in \mathcal{Z}^{+(i)} \end{cases}$$

*Proof:* First we write down the primal formulation, and assign the Lagrange multipliers:

$$f^{\star} = \min_{\boldsymbol{x}, \hat{\boldsymbol{x}}, \mathbf{z}} f(x)$$

$$\text{s.t.} \quad f(\boldsymbol{x}) = \boldsymbol{x}^{(L)}; \quad \boldsymbol{x}_0 - \epsilon \leq \boldsymbol{x} \leq \boldsymbol{x}_0 + \epsilon; \tag{4}$$

$$\boldsymbol{x}^{(i)} = \mathbf{W}^{(i)} \hat{\boldsymbol{x}}^{(i-1)} + \mathbf{b}^{(i)}; \quad i \in [L], \qquad\qquad \Rightarrow \boldsymbol{\nu}^{(i)} \in \mathbb{R}^{d_i} \tag{5}$$

For $i \in [L-1]$:

$$\hat{x}_j^{(i)} \geq 0; \; j \in \mathcal{I}^{(i)} \qquad\qquad \Rightarrow \mu_j^{(i)} \in \mathbb{R}_+, j \in \mathcal{I}^{(i)} \setminus \mathcal{Z}^{-(i)} \tag{8}$$

$$\hat{x}_j^{(i)} \geq x_j^{(i)}; \; j \in \mathcal{I}^{(i)} \qquad\qquad \Rightarrow \tau_j^{(i)} \in \mathbb{R}_+, j \in \mathcal{I}^{(i)} \setminus \mathcal{Z}^{+(i)} \tag{9}$$

$$\hat{x}_j^{(i)} \leq u_j^{(i)} z_j^{(i)}; \; j \in \mathcal{I}^{(i)} \qquad\qquad \Rightarrow \gamma_j^{(i)} \in \mathbb{R}_+, j \in \mathcal{I}^{(i)} \setminus \mathcal{Z}^{(i)} \tag{10}$$

$$\hat{x}_j^{(i)} \leq x_j^{(i)} - l_j^{(i)}(1 - z_j^{(i)}); \; j \in \mathcal{I}^{(i)} \qquad\qquad \Rightarrow \pi_j^{(i)} \in \mathbb{R}_+, j \in \mathcal{I}^{(i)} \setminus \mathcal{Z}^{(i)} \tag{11}$$

$$\hat{x}_j^{(i)} = x_j^{(i)}; \; j \in \mathcal{I}^{+(i)} \tag{13}$$

$$\hat{x}_j^{(i)} = 0; \; j \in \mathcal{I}^{-(i)} \tag{14}$$

$$0 \leq z_j^{(i)} \leq 1; j \in \mathcal{I}^{(i)} \setminus \mathcal{Z}^{(i)} \tag{12}$$

$$z_j^{(i)} = 1, \forall j \in \mathcal{Z}^{+(i)}; \quad z_j^{(i)} = 0, \forall j \in \mathcal{Z}^{-(i)} \tag{20}$$

$$\sum_{i=1}^{L-1} \left(\boldsymbol{H}^{(i)} \boldsymbol{x}^{(i)} + \boldsymbol{G}^{(i)} \hat{\boldsymbol{x}}^{(i)} + \boldsymbol{Q}^{(i)} \mathbf{z}^{(i)}\right) \leq \boldsymbol{d} \qquad\qquad \Rightarrow \boldsymbol{\beta} \in \mathbb{R}_+^N \tag{2}$$

Note that for some constraints, their dual variables are not created because they are trivial to handle in the steps. And from Lemma A.1, some unstable neuron constraints 8, 9, 10, 11 will be eliminated during BaB with split on $z$, i.e. when $z_j^{(i)} = 1$, the unstable neuron constraints 8 will be reduced to

$\hat{x}_j^{(i)} = x_j^{(i)} \geq 0 \Rightarrow \mu_j^{(i)} \in \mathbb{R}_+$, other dual variables for constraints 9, 10, 11 will not be generated, and when $z_j^{(i)} = 0$, only $\tau_j^{(i)} \in \mathbb{R}_+$ will be generated. Rearrange the equation and swap the min and max (strong duality) gives us:

$$
\begin{aligned}
f^\star = \max_{\boldsymbol{\nu},\boldsymbol{\mu},\boldsymbol{\tau},\boldsymbol{\gamma},\boldsymbol{\pi},\boldsymbol{\beta}} \min_{\boldsymbol{x},\hat{\boldsymbol{x}},\boldsymbol{z}} \ & (\boldsymbol{\nu}^{(L)} + 1)\boldsymbol{x}^{(L)} - \boldsymbol{\nu}^{(1)\top}\mathbf{W}^{(1)}\hat{\boldsymbol{x}}^{(0)} \\
& + \sum_{i=1}^{L-1}\sum_{j\in\mathcal{I}^{+(i)}} \left(\nu_j^{(i)} + \boldsymbol{\beta}^\top \boldsymbol{H}_{:,j}^{(i)} - \boldsymbol{\nu}^{(i+1)\top}\mathbf{W}_{:,j}^{(i+1)} + \boldsymbol{\beta}^\top \boldsymbol{G}_{:,j}^{(i)}\right) x_j^{(i)} \\
& + \sum_{i=1}^{L-1}\sum_{j\in\mathcal{Z}^{+(i)}} \left(\nu_j^{(i)} + \boldsymbol{\beta}^\top \boldsymbol{H}_{:,j}^{(i)} - \boldsymbol{\nu}^{(i+1)\top}\mathbf{W}_{:,j}^{(i+1)} + \boldsymbol{\beta}^\top \boldsymbol{G}_{:,j}^{(i)} - \mu_j^{(i)}\right) x_j^{(i)} \\
& + \sum_{i=1}^{L-1}\sum_{j\in\mathcal{I}^{-(i)}} \left(\nu_j^{(i)} + \boldsymbol{\beta}^\top \boldsymbol{H}_{:,j}^{(i)}\right) x_j^{(i)} + \sum_{i=1}^{L-1}\sum_{j\in\mathcal{Z}^{-(i)}} \left(\nu_j^{(i)} + \boldsymbol{\beta}^\top \boldsymbol{H}_{:,j}^{(i)} + \tau_j^{(i)}\right) x_j^{(i)} \\
& + \sum_{i=1}^{L-1}\sum_{j\in\mathcal{I}^{(i)}} \Big[ \left(\nu_j^{(i)} + \boldsymbol{\beta}^\top \boldsymbol{H}_{:,j}^{(i)} + \tau_j^{(i)} - \pi_j^{(i)}\right) x_j^{(i)} \\
& \quad + \left(-\boldsymbol{\nu}^{(i)\top}\mathbf{W}_{:,j}^{(i)} - \mu_j^{(i)} - \tau_j^{(i)} + \gamma_j^{(i)} + \pi_j^{(i)} + \boldsymbol{\beta}^\top \boldsymbol{G}_{:,j}^{(i)}\right) \hat{x}_j^{(i)} \\
& \quad + \left(-u_j^{(i)}\gamma_j^{(i)} - l_j^{(i)}\pi_j^{(i)} + \boldsymbol{\beta}^\top \boldsymbol{Q}_{:,j}^{(i)}\right) z_j^{(i)} \Big] \\
& - \sum_{i=1}^{L} \boldsymbol{\nu}^{(i)\top}\mathbf{b}^{(i)} + \sum_{i=1}^{L-1}\sum_{j\in\mathcal{I}^{(i)}\setminus\mathcal{Z}^{(i)}} \pi_j^{(i)} l_j^{(i)} - \boldsymbol{\beta}^\top \boldsymbol{d}
\end{aligned}
$$

s.t. $\quad \boldsymbol{x}_0 - \epsilon \leq \boldsymbol{x} \leq \boldsymbol{x}_0 + \epsilon; \quad 0 \leq z_j^{(i)} \leq 1, j \in \mathcal{I}^{(i)} \setminus \mathcal{Z}^{(i)} \quad z_j^{(i)} = 1, j \in \mathcal{Z}^{+(i)} \quad z_j^{(i)} = 0, j \in \mathcal{Z}^{-(i)}$

$\qquad \boldsymbol{\mu} \geq 0; \quad \boldsymbol{\tau} \geq 0; \quad \boldsymbol{\gamma} \geq 0; \quad \boldsymbol{\pi} \geq 0; \quad \boldsymbol{\beta} \geq 0$

Here $\mathbf{W}_{:,j}^{(i+1)}$ denotes the $j$-th column of $\mathbf{W}^{(i+1)}$. Note that for the term involving $j \in \mathcal{I}^{+(i)} \cup \mathcal{Z}^{+(i)}$ we have replaced $\hat{\boldsymbol{x}}^{(i)}$ with $\boldsymbol{x}^{(i)}$ to obtain the above equation. For the term $x_j^{(i)}, j \in \mathcal{I}^{-(i)} \cup \mathcal{Z}^{-(i)}$ it is always 0 so it does not appear. Then solving the inner minimization gives us the dual formulation:

$$
f^\star = \max_{\boldsymbol{\nu},\boldsymbol{\mu},\boldsymbol{\tau},\boldsymbol{\gamma},\boldsymbol{\pi},\boldsymbol{\beta}\geq 0} -\epsilon\|\boldsymbol{\nu}^{(1)\top}\mathbf{W}^{(1)}\boldsymbol{x}_0\|_1 - \sum_{i=1}^{L} \boldsymbol{\nu}^{(i)\top}\mathbf{b}^{(i)} - \boldsymbol{\beta}^\top \boldsymbol{d}
$$

$$
+ \sum_{i=1}^{L-1}\sum_{j\in\mathcal{I}^{(i)}} h_j^{(i)} \tag{21}
$$

$$
\text{s.t. } \boldsymbol{\nu}^{(L)} = -1 \tag{22}
$$

$$
\nu_j^{(i)} = \boldsymbol{\nu}^{(i+1)\top}\mathbf{W}_{:,j}^{(i+1)} - \boldsymbol{\beta}^\top(\boldsymbol{H}_{:,j}^{(i)} + \boldsymbol{G}_{:,j}^{(i)}), \quad \text{for } j \in \mathcal{I}^{+(i)}, i \in [L-1] \tag{23}
$$

$$
\nu_j^{(i)} = \boldsymbol{\nu}^{(i+1)\top}\mathbf{W}_{:,j}^{(i+1)} - \boldsymbol{\beta}^\top(\boldsymbol{H}_{:,j}^{(i)} + \boldsymbol{G}_{:,j}^{(i)}) + \mu_j^{(i)}, \quad \text{for } j \in \mathcal{Z}^{+(i)}, i \in [L-1] \tag{24}
$$

$$
\nu_j^{(i)} = -\boldsymbol{\beta}^\top \boldsymbol{H}_{:,j}^{(i)}, \quad \text{for } j \in \mathcal{I}^{-(i)}, i \in [L-1] \tag{25}
$$

$$
\nu_j^{(i)} = -\boldsymbol{\beta}^\top \boldsymbol{H}_{:,j}^{(i)} - \tau_j^{(i)}, \quad \text{for } j \in \mathcal{Z}^{-(i)}, i \in [L-1] \tag{26}
$$

$$
\nu_j^{(i)} = \pi_j^{(i)} - \tau_j^{(i)} - \boldsymbol{\beta}^\top \boldsymbol{H}_{:,j}^{(i)} \quad \text{for } j \in \mathcal{I}^{(i)} \setminus \mathcal{Z}^{(i)}, i \in [L-1] \tag{27}
$$

$$
\left(\pi_j^{(i)} + \gamma_j^{(i)}\right) - \left(\mu_j^{(i)} + \tau_j^{(i)}\right) = \boldsymbol{\nu}^{(i+1)\top}\mathbf{W}_{:,j}^{(i+1)} - \boldsymbol{\beta}^\top \boldsymbol{G}_{:,j}^{(i)}, \quad \text{for } j \in \mathcal{I}^{(i)} \setminus \mathcal{Z}^{(i)}, i \in [L-1] \tag{28}
$$

where $h_j^{(i)}$ in the objective 21 is:

$$h_j^{(i)} = \min_{z_j^{(i)}}\{\pi_j^{(i)}l_j^{(i)} + (-u_j^{(i)}\gamma_j^{(i)} - l_j^{(i)}\pi_j^{(i)} + \boldsymbol{\beta}^\top \boldsymbol{Q}_{:,j}^{(i)})z_j^{(i)}\} \tag{29}$$

$$= \begin{cases} \boldsymbol{\beta}^\top \boldsymbol{Q}_{:,j}^{(i)} & \text{if } j \in \mathcal{Z}^{+(i)} \\ 0 & \text{if } j \in \mathcal{Z}^{-(i)} \\ \pi_j^{(i)}l_j^{(i)} - \text{ReLU}(u_j^{(i)}\gamma_j^{(i)} + l_j^{(i)}\pi_j^{(i)} - \boldsymbol{\beta}^\top \boldsymbol{Q}_{:,j}^{(i)}) & \text{if } j \in \mathcal{I}^{(i)} \setminus \mathcal{Z}^{(i)} \end{cases} \tag{30}$$

The $\text{ReLU}(\cdot)$ term comes from minimizing over $z_j^{(i)}$ with the constraint $0 \le z_j^{(i)} \le 1$. And when splitting on $z \in \mathcal{Z}^{(i)}$, as we discussed above, based on Lemma A.1, there will be no $\pi_j^{(i)}, \gamma_j^{(i)}$ assigned for the constraints. The $\|\cdot\|_1$ comes from the dual norm form minimizing over the infinite norm of the input $\boldsymbol{x}$ with the constraint $\boldsymbol{x}_0 - \epsilon \le \boldsymbol{x} \le \boldsymbol{x}_0 + \epsilon$.

Thus, the rest of the proof follows from [61, Lemma A.1, Theorem 3.1], which completes the proof of Theorem 3.2.

### B.3 Proof of Corollary 3.3

*Proof.* Let us consider any assignment of the variables $\{z_i\}$ where $z_i \in \{0,1\}$ that satisfies cut 3.1. We aim to show that this assignment also satisfies cut 3.2.

Since $\mathcal{Z}_1^+ \subset \mathcal{Z}_2^+$, there exists at least one neuron $k$ such that $k \in \mathcal{Z}_2^+$ but $k \notin \mathcal{Z}_1^+$. Let $\mathcal{Z}_+^{(\Delta)} = \mathcal{Z}_2^+ \setminus \mathcal{Z}_1^+$ denote the additional neurons in the larger cut 3.2. Similarly, define $\mathcal{Z}_-^{(\Delta)} = \mathcal{Z}_2^- \setminus \mathcal{Z}_1^-$.

In cut 3.2, the left-hand side (LHS) can be expressed as:

$$\text{LHS}_{3.2} = \left( \sum_{i \in \mathcal{Z}_1^+} z_i + \sum_{i \in \mathcal{Z}_+^{(\Delta)}} z_i \right) - \left( \sum_{i \in \mathcal{Z}_1^-} z_i + \sum_{i \in \mathcal{Z}_-^{(\Delta)}} z_i \right)$$

Similarly, the right-hand side (RHS) of cut 3.2 is:

$$\text{RHS}_{3.2} = |\mathcal{Z}_1^+| + |\mathcal{Z}_+^{(\Delta)}| - 1$$

Subtracting Eq. 3.1 from cut 3.2, we get:

$$\text{LHS}_{3.2} - \left( \sum_{i \in \mathcal{Z}_1^+} z_i - \sum_{i \in \mathcal{Z}_1^-} z_i \right) = \left( \sum_{i \in \mathcal{Z}_+^{(\Delta)}} z_i - \sum_{i \in \mathcal{Z}_-^{(\Delta)}} z_i \right)$$

$$\text{RHS}_{3.2} - (|\mathcal{Z}_1^+| - 1) = |\mathcal{Z}_+^{(\Delta)}|$$

Therefore, the difference between the LHS and RHS of the two cuts is:

$$\left( \sum_{i \in \mathcal{Z}_+^{(\Delta)}} z_i - \sum_{i \in \mathcal{Z}_-^{(\Delta)}} z_i \right) \le |\mathcal{Z}_+^{(\Delta)}|$$

Since each $z_i \in \{0,1\}$, the maximum value of $\sum_{i \in \mathcal{Z}_+^{(\Delta)}} z_i$ is $|\mathcal{Z}_+^{(\Delta)}|$, and the minimum value of $\sum_{i \in \mathcal{Z}_-^{(\Delta)}} z_i$ is 0. Thus, the maximum possible value of the LHS difference is $|\mathcal{Z}_+^{(\Delta)}|$, which equals the RHS difference.

Therefore, regardless of the values of $z_i$ for $i \in \mathcal{Z}_+^{(\Delta)} \cup \mathcal{Z}_-^{(\Delta)}$, the inequality:

$$\left( \sum_{i \in \mathcal{Z}_+^{(\Delta)}} z_i - \sum_{i \in \mathcal{Z}_-^{(\Delta)}} z_i \right) \le |\mathcal{Z}_+^{(\Delta)}|$$

is always satisfied. This means that satisfying cut 3.1 ensures that cut 3.2 is also satisfied.

Conversely, consider an assignment where cut 3.2 is satisfied but cut 3.1 is violated. This can happen if the additional variables in $\mathcal{Z}_+^{(\Delta)}$ and $\mathcal{Z}_-^{(\Delta)}$ compensate for the violation in the original variables. For instance, setting $z_i = 1$ for all $i \in \mathcal{Z}_+^{(\Delta)}$ and $z_i = 0$ for all $i \in \mathcal{Z}_-^{(\Delta)}$ can decrease the LHS of cut 3.2, making it easier to satisfy even if cut 3.1 is not satisfied.

Therefore, the feasible region defined by cut 3.1 is a subset of the feasible region defined by cut 3.2, confirming that cut 3.1 is stronger.

□

## C Experiments

### C.1 Experiment Settings

Our experiments are conducted on a server with an Intel Xeon 8468 Sapphire CPU, one NVIDIA H100 GPU (96 GB GPU memory), and 480 GB CPU memory. Our implementation is based on the open-source $\alpha,\beta$-CROWN verifier[1] with cutting plane related code added. All experiments use 24 CPU cores and 1 GPU. The MIP cuts are acquired by the `cplex` [31] solver (version 22.1.0.0).

We use the Adam optimizer [33] to solve both $\alpha, \beta, \mu, \tau$. For the SDP-FO benchmarks, we optimize those parameters for 20 iterations with a learning rate of 0.1 for $\alpha$ and 0.02 for $\beta, \mu, \tau$. We decay the learning rates with a factor of 0.98 per iteration. The timeout is 200s per instance. For the VNN-COMP benchmarks, we use the same configuration as $\alpha,\beta$-CROWN used in the respective competition and the same timeouts. During constraint strengthening, we set **drop_percentage = 50%**. We only perform one round of strengthening, with no recursive strengthening attempts. We perform constraint strengthening for the first 40 BaB iterations. For multi-tree search, we perform 5 branching iterations, where we each time pick the current best 50 domains and split them to generate 400 new sub-domains.

### C.2 Ablation Studies: Number of Cuts and Branch Visited Analysis

The ablation studies provide insights into the impact of different components of the BICCOS algorithm on the verification process, focusing on the average number of branches (domains) explored and the number of cuts generated, as shown in Table 4. A lower number of branches typically indicates a more efficient search process, leading to computational savings and faster verification times.

Across the benchmarks, we observe that the BICCOS configurations, particularly BICCOS (auto), often explore fewer branches compared to the baseline $\beta$-CROWN and GCP-CROWN methods. For instance, on the CIFAR CNN-A-Adv model, BICCOS (auto) reduces the average number of branches from 63,809.77 (for $\beta$-CROWN) and 28,558.70 (for GCP-CROWN) down to 9,902.96. This reduction is attributed to the cutting planes introduced by the BICCOS algorithm, which tighten the relaxations and more effectively prune suboptimal regions of the search space.

The multi-tree search (MTS) strategy, which explores multiple branch-and-bound trees in parallel, demonstrates its effectiveness in improving verification efficiency. While MTS may increase the total number of branches explored—since domains from multiple trees are counted—the exploration within each tree is optimized, leading to faster convergence and reduced overall computation time. We also design an adaptive BICCOS configuration (BICCOS (auto)), which automatically enables MTS and/or Mixed Integer Programming (MIP)-based cuts based on the neural network and verification problem size. BICCOS (auto) achieves the highest verified accuracies across models and is used as the default option of the verifier when BICCOS is enabled.

---

[1]`https://github.com/huanzhang12/alpha-beta-CROWN`

Table 4: Ablation Studies on avg. # of cuts and branches visited analysis for **BaB verified** (hard) instances on different BICCOS components.

| Dataset | Model $\epsilon = 0.3$ and $\epsilon = 2/255$ | $\beta$-CROWN [54] domain visited # | cut # | GCP-CROWN [61] domain visited # | cut # | BICCOS (base) domain visited # | cut # | BICCOS (with MTS) domain visited # | cut # | BICCOS (auto) domain visited # | cut # |
|---|---|---|---|---|---|---|---|---|---|---|---|
| MNIST | CNN-A-Adv | 3145.14 | 0 | 5469.28 | 492.70 | 3330.67 | 113.36 | 3543.39 | 132.98 | 2857.88 | 611.50 |
| CIFAR | CNN-A-Adv | 63809.77 | 0 | 28558.70 | 353.49 | 18613.68 | 169.00 | 20134.27 | 159.64 | 9902.96 | 506.88 |
| | CNN-A-Adv-4 | 14873.88 | 0 | 15313.58 | 565.27 | 7835.50 | 131.08 | 7470.90 | 132.09 | 6730.61 | 691.00 |
| | CNN-A-Mix | 24965.58 | 0 | 27779.75 | 436.77 | 11832.11 | 118.06 | 12238.09 | 121.25 | 15775.58 | 597.56 |
| | CNN-A-Mix-4 | 1565.91 | 0 | 18271.2 | 603 | 26608.61 | 154.25 | 8253.62 | 143.62 | 10059.78 | 752.33 |
| | CNN-B-Adv | 18106.46 | 0 | 17309.63 | 273.78 | 9803.96 | 163.06 | 9012.67 | 198.54 | 12782.82 | 332.17 |
| | CNN-B-Adv-4 | 20987.40 | 0 | 37661.92 | 267.71 | 7545.83 | 91.2 | 11585.87 | 90.90 | 17886.02 | 351.85 |
| oval21 | | 36299.41 | 0 | 13182.31 | 5635.37 | 10589.31 | 147.94 | 7377.77 | 134.83 | 20937.95 | 5718.17 |
| cifar100-2024 | | 1886.07 | 0 | 1884.02 | 0 | 1465.01 | 128.75 | 892.11 | 147.92 | 1216.49 | 145.75 |
| tinyimagenet-2024 | | 906.69 | 0 | 809 | 0 | 902.97 | 134.99 | 896.51 | 169.41 | 874.05 | 181.53 |

*We run our BICCOS in different ablation studies with a shorter 200s timeout for all models and compare it to $\beta$-CROWN and GCP-CROWN, it achieves better verified accuracy than all other baselines.

However, it is worth noting that on certain benchmarks, such as the CIFAR CNN-A-Mix-4 model, the number of branches explored by different BICCOS configurations does not always decrease compared to the baselines. In some cases, the base BICCOS version explores more branches than $\beta$-CROWN. This could be attributed to the models' inherent complexity and the nature of the verification problem, where the benefits of the cutting planes and MTS may be less pronounced.

Overall, the analysis of the average number of branches in the ablation studies highlights the effectiveness of the BICCOS algorithm and its components in improving the efficiency of the verification process. The combination of cutting planes, multi-tree search, and other optimizations enables BICCOS to achieve high verified accuracy while often exploring fewer branches compared to baseline methods, ultimately leading to computational savings and faster verification times.

### C.3 Comparison with MIP based Verifier

To evaluate the performance of the BICCOS cuts in MIP-based solvers, Fig 3 shows comparison with Venus2 [9, 34]. We emphasize on making a fair comparison on the strengths of cuts. Since Venus2 uses an MILP solver to process its cuts, in these experiments we do not use the efficient GCP-CROWN solver. Instead, we also use an MILP solver to handle the BICCOS cuts we found. This ensures that the speedup we achieve is not coming from the GPU-accelerated GCP-CROWN solver. Since our cut generation relies on the process with BaB, we first run BICCOS to get the cuts, and then import the cuts into the MILP solver.

We note that Venus uses branching over ReLU activations to create multiple strengthened MILP problems. On the other hand, we only create one single MILP and do not perform additional branching. Therefore, our MILP formulation is weaker. The fact that we can outperform Venus2 anyway underlines the strength of the generated cuts by BICCOS.

### C.4 Overhead Analysis

We note that the multi-tree search (MTS) and cut strengthening procedure incur additional overheads compared to the baseline verifiers shown as Fig 4. The MTS has about 2 - 10 seconds overhead in the benchmarks we evaluated, which may increase the verification time of very easy instances that can be sequentially immediately verified in branch and bound. The BICCOS cut strengthening is called during every branch-and-bound iteration, but its accumulative cost is about 3 - 20 seconds for hard instances in each benchmark, which run a few hundred branch-and-bound iterations. We note that typically, the performance of the verifier is gauged by the number of verified instances within a fixed timeout threshold, thus when the threshold is too low, the added cutting planes may not have time to show their performance gains due to the lack of sufficient time for the branch-and-bound procedure.

We note that such a shortcoming is also shared with other verifiers which require additional steps for bound tightening; for example, the GCP-CROWN verifiers have the extra overhead of a few seconds of calling and retrieving results from the MIP solver, even when the MIP solver is running in parallel with the verifier. The final verified accuracy can justify whether the overhead is worth paying.

## D More Related Works

Furthermore, nogood learning and conflict analysis techniques from constraint programming and SAT solving have been adapted to neural network verification to improve solver efficiency [18, 45, 26, 19,

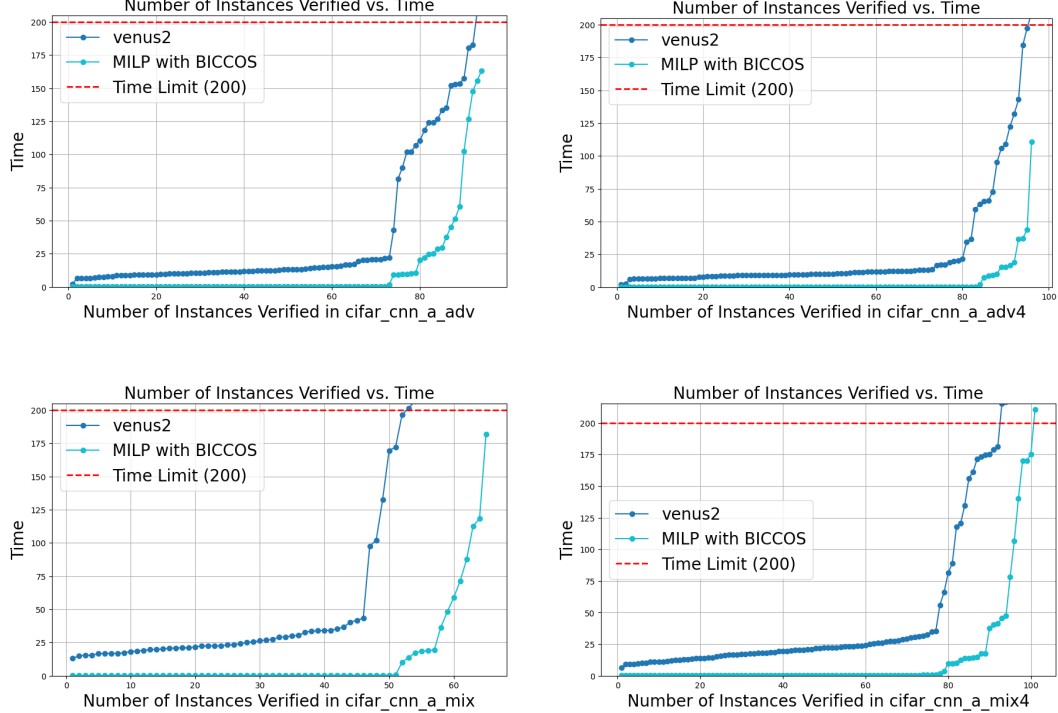

Figure 3: Comparison of Venus2 and MILP with BICCOS. For a fair comparison, we do not use GPU-accelerated bound propagation but use a MILP solver (same as in Venus2) to solve the verification problem with our BICCOS cuts. In all 4 benchmarks, MILP with BICCOS cuts is faster than Venus (MILP with their proposed cuts), illustrating the effectiveness. Note that Venus2 can hardly scale to larger models presented in our paper, such as those on `cifar100` and `tinyimagenet` datasets.

11]. These methods record conflicts encountered during the search—known as nogoods—to prevent revisiting the same conflicting states, effectively pruning the search space. While nogood learning can reduce redundant exploration, it typically operates on discrete abstractions of neural networks, representing ReLU activations as boolean variables. This abstraction limits their ability to leverage the properties of neural network activations. In contrast, our approach not only identifies conflicts but also translates them into cutting planes within the continuous relaxation of the verification problem. This allows us to tighten the bounds for unexplored subproblems, enhancing the overall efficiency of the verification process.

Mixed Integer Programming (MIP) solvers, such as `cplex` [31] and `gurobi` [27], have also been applied to neural network verification by formulating the problem as a mixed-integer linear program [6]. These solvers inherently utilize branch-and-bound algorithms and incorporate sophisticated cutting plane generation techniques to tighten the feasible region. However, in practice, applying general-purpose MIP solvers to large-scale neural networks presents significant challenges. For instance, in our experiments with benchmarks like CIFAR100 and Tiny-ImageNet—where network sizes range from 5.4 million to over 30 million parameters—`cplex` was unable to solve the initial LP relaxation within the 200-second timeout threshold. As a result, the branch-and-bound process and cut generation did not commence, underscoring the limitations of using off-the-shelf MIP solvers for large-scale neural network verification tasks.

In terms of cutting plane methods, prior work has explored integrating cuts derived from convex hull constraints within MIP formulations of neural networks [2, 47, 52, 30, 61, 35]. For example, [2] propose cutting planes that tighten the relaxation of ReLU constraints in MILP formulations, improving solution quality. While these methods can enhance optimization, they often incur significant computational overhead in generating and integrating cuts and rely heavily on the capabilities of the underlying MIP solver. In contrast, our approach develops new cutting planes that can be efficiently constructed and strengthened during the branch-and-bound process, leveraging the problem's structure to achieve better scalability and performance.

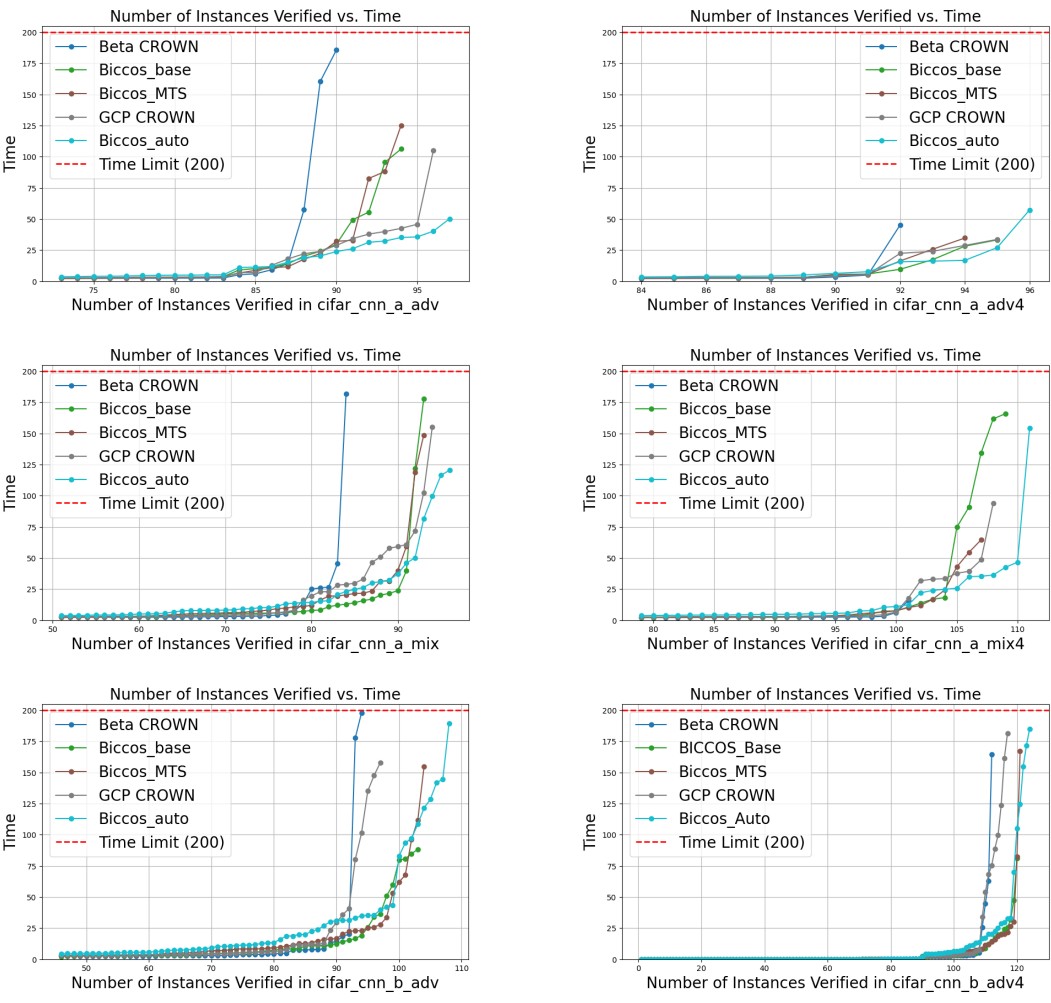

Figure 4: Slow down comparison on easier properties on BICCOS (base), BICCOS (MTS), GCP-CROWN with MIP cuts and BICCOS (auto) . We plot the number of solved instances versus verification time for BICCOS and $\beta$-CROWN (baseline). For easy instances that can be verified within a few seconds (bottom parts of the figures), the increase of verification time with BICCOS is negligible.

Our work bridges the gap between conflict analysis methods like nogood learning and cutting plane techniques by integrating conflict-derived constraints into the continuous relaxation of the verification problem. This integration allows for both effective pruning of the search space and tightening of the problem relaxation, leading to improved verification efficiency. By tailoring our approach to the specifics of neural network verification, we address the limitations observed in general-purpose MIP solvers when applied to large-scale networks, offering a scalable and efficient solution.

# E Limitations

The approach relies on the piecewise linear nature of ReLU networks (same as most other papers in BaB), and its applicability to other types of neural architectures requires further extension such as those in [48]. Additionally, the effectiveness of the generated cutting planes may vary depending on the specific structure and complexity of the neural network being verified, and there may be cases where the algorithm's performance gains are less pronounced.

