# OpenReview forum: "Scalable Neural Network Verification with Branch-and-bound Inferred Cutting Planes"
_NeurIPS.cc/2024/Conference — NeurIPS 2024 poster_

### Official Review · Reviewer_iWFh · 2024-07-06

**Soundness:** 3
**Presentation:** 2
**Contribution:** 3
**Rating:** 6
**Confidence:** 5

**Summary:**

Following the discussion with the authors, I am increasing my score to weak accept.
----

This paper presents a new branch-and-bound-based verification algorithm for neural networks with ReLU activation. The main idea is to produce additional constraints to the verification problem by identifying small combinations of neuronal states which would not lead to an adversarial input, so that these combinations are not explored again in another part of the branch-and-bound tree. Because smaller sets are always better, the authors develop an algorithm for reducing the number of neurons involved in such combinations. They also leverage multiple branch-and-bound trees explored in breadth-first search as a means of producing constraints involving fewer neurons.

**Strengths:**

The ideas presented by the authors are intuitively reasonable and unsurprisingly effective as a consequence. In fact, they have been studied in other contexts with other names. Most notably, the idea of producing a constraint based on a path of the branch-and-bound tree is commonly known as nogood learning (see [1] for an early use in constraint programming, such as with the concept of a conflict set in page 9; and [2] for the first application to mixed-integer programming). The canonical application has been to identify infeasible nodes, but it is not a big stretch to apply the same idea for when the objective function is likely suboptimal (positive objective function when what we need is a negative value). Moreover, the idea of exploring multiple branch-and-bound trees in parallel is not far from solver restarts [3], which have been widely used in constraint programming and subsequently adopted in mixed-integer programming as well.

All in all, this is a paper that speaks for itself because the right ideas were used and better numbers were obtained. Issues come next.

[1] https://ftp.cs.ucla.edu/pub/stat_ser/r77-II-reprint.pdf

[2] https://www.cs.cmu.edu/~sandholm/nogoodsForMip.techReport06.pdf

[3] http://www.cs.cornell.edu/selman/papers/pdf/98.aaai.boost.pdf

**Weaknesses:**

One problematic omission in this paper is that nowhere it is said that MIP solvers also use branch-and-bound. Otherwise, it sounds as if MIP solvers are some sort of black box magic that tries to solve problems in some inefficient way. There is also some broad generalizations such as "MIP solver may not return any cuts before timeout" (149): Which solver? When? Under which conditions? All that a generic MIP solver needs to generate a Gomory cut is the solution of the LP relaxation and the corresponding tableau.

The fact that branches are defined in terms of neuron outputs is also problematic because there is an overlap between the two resulting subproblems when x=0. The effectiveness of branch-and-bound comes from partitioning the feasible set, and moreover from slicing off the parts of the LP relaxation that are not feasible according to the MIP formulation (such as when z=0 is one subproblem, z=1 is another subproblem, and 0<z<1 gets thrown away). Because your implementation is extending someone else's work, I believe that this might be a mismatch between the code and what the paper describes. Since the z variables are nevertheless a way to parameterize the LP problems to be solved along the way, there would be no problem in defining the constraints directly in terms of z instead of x. In fact, this is exactly what you do when you finally formalize the cut that you are using.

When it gets to Proposition 3.1, the statement is unnecessarily complex: why define a diagonal matrix only to multiply it by a vector of ones immediately after? In terms of notation, it doesn't make sense to have output variables indexing which neurons should be removed. Instead, you could use only the corresponding indices. Hence, Z = {1, 3} instead of Z = {x_1, x_3}. Moreover, this is a commonly known cutting plane in MIP, perhaps first introduced in 1972 by [4] and at this point widely known and applied in the MIP community. It would have been better to use the conventional notation $\sum_{i \in Z^+} z_i - \sum_{i \in Z^-} z_i \leq |Z^+| - 1$ directly in the paper rather than using it in the appendix and keeping a less readable version in the paper.

[4] https://epubs.siam.org/doi/10.1137/0123007

Algorithm 1 relies on a threshold, but values or experiments to find the right one are not mentioned anywhere.

The paper also talks very briefly about prior work, to the point that it sounds as if specialized cutting planes for MIP problems involving neural networks were never proposed before. However, your reference 1 (Anderson et al.) does exactly that. The same is true about [5-6], and a broader perspective about this area can be found in Section 4 of the survey [7].

[5] https://arxiv.org/abs/1810.03370

[6] https://arxiv.org/abs/2102.04373

[7] https://arxiv.org/abs/2305.00241

Other major comments about the writing follow below.

A) The "The NN Verification Problem" paragraph in Section 2 is not very precise because it confuses values with variables and do not properly qualify functions when values are dependent on inputs. What follows is a possible correction to that. In line 84, replace "scalar values" with "scalar variables". In line 85, add "the values of" after "for". In line 86, replace ", when they depend on a specific x" with "for a given input x". In line 87, add to "that limit [the post-activation values of] individual neurons".

B) The text talks about an example in which you no longer want to see both neurons 1 and 3 inactive at the same time, but the example in Figure 1 shows neuron 3 active.

Other minor comments about writing follow below.

10-11: "cutting planes constraints": this is not an adequate expression (all constraints define a plane); use either cutting planes, valid inequalities, constraints, or cuts.

21-22: First sentence is a strong statement with no reference provided and never discussed again.

32: "statuses"   ->   states (again in 53)

67: "discussed"   ->   discuss (not past tense)

98: "> 0"   ->   < 0?

98: "pro[p]erty"

125: "status"   ->   state

139: remove "BIC"

149: "and [the] MIP solver"

162: "positive regime": active regime?

163: "in [the] LP formulation"

Authors repeated twice in reference 11.

References 13 and 14 are the same.

Use {CDCL} instead of CDCL in reference 28 and {SAT} instead of SAT in reference 30 to keep the casing in the output, since otherwise what you get is cdcl and sat.

**Questions:**

1) In light of the comments about nogood learning and restarts, can the authors please reframe their contribution in the paper in terms of the existing related work?

2) Can you please qualify your discussion about MIP solvers by being more precise in what they can and cannot do?

3) Are you branching on the output of the neurons (x) , or on the state of the neurons (z)?

4) Can you simplify Proposition 3.1?

5) Can you please describe what thresholds were used for Algorithm 1, and how they were obtained?

6) Can you please rectify the discussion about cutting planes for neural networks by acknowledging prior work on this?

7) Can you please comment on incorporating the major and minor corrections to writing?

8) In lines 145-146, what do you mean by "[49] solved these cuts using GPU-accelerated bound propagation"? What is the concept of solving a cut? Do you mean generating the cut? Do you mean solving the LP relaxation after the cuts are included?

**Limitations:**

The authors were very upfront about limitations to their work.

---

> ### Author Rebuttal · Authors · 2024-08-07
>
> We thank the reviewer for the constructive comments and valuable questions. We hope the reviewer can reevaluate our paper based on our response below:
>
> - Q1: Reframe the contribution in the paper in terms of the existing related work about nogood learning and restarts.
>
>    Thank you for your valuable input, which allows us to view our research from a broader perspective. Other NN verification papers, such as beta-CROWN, have utilized branch-and-bound techniques specifically tailored to NN verification. While branch-and-bound is a general technique, its adaptation to the unique challenges and requirements of NN verification is what distinguishes our work. We will make sure to cite these works appropriately and discuss their relevance to our context.
>
> - Q2: Discussion about MIP solvers more precisely.
>
>     Thank you for your comments highlighting the need for a more detailed discussion regarding MIP solvers. We will clarify in our revision that MIP solvers, including CPLEX which we primarily used, employ branch-and-bound algorithms as a fundamental part of their operation. We found that in many of these large-scale benchmarks, the MIP solver fails to solve the initial (root) LP relaxation for the verification problem within the timeout threshold, so branch-and-bound has never started, and also no effective cuts can be generated. For instance, in the cifar100 benchmark with networks size between 5.4-31.6 M and tiny-imagnet with a network size of 14.4 M, CPLEX was unable to generate any cuts within 200 seconds (timeout threshold for this benchmark). These examples will illustrate the limitations more concretely and contextualize the generic statements we previously made, which we will add this discussion to our revised paper.
>
> - Q3: Clarification of  branching on the output of $x$ or $z$.
>
>     By definition (line 100) of $z$, $z=0$ is equivalent to $x \leq 0$, and $z=1$ is equivalent to $x \geq 0$. This is encoded in Appendix A1, Equation 12 and 13: By inserting z=0 (z=1), x is restricted to the respective non-negative (non-positive) domain. Therefore, branching over $x \leq 0$ vs $x \geq 0$ is equivalent to branching over $z=0$ vs $z=1$. Note that technically we cannot use strict inequality like $x > 0$ or $x < 0$, so the $x=0$ case is needed in both branches. The equvalence of the two formulations has been used in prior work such as GCP-CROWN.
>
> - Q4: simplify Proposition 3.1
>
>    We will remove the diagonal matrix and simplify the notation. Additionally, we acknowledge that this cutting plane is a commonly known concept in MIP, first introduced in 1972 by [4]. Here is the revised cut expression:
>
>    $$\sum_{i \in \mathcal{Z}^{+}} z_i - \sum_{i \in \mathcal{Z}^{-}} z_i \leq |\mathcal{Z}^{+}| - 1$$
>
> - Q5: Thresholds used for Algorithm 1 and how do we obtain it
>
>     The specific threshold value and its justification were detailed in Section A.3.1. The threshold, `drop_percentage`=50% was determined based on preliminary experiments during code development and empirical observations. We conducted multiple trials to assess the impact of different threshold values on the performance and effectiveness of the algorithm. Through these experiments, we observed that a 50% threshold provided a balanced trade-off between pruning weak constraints and maintaining a robust constraint set. We will ensure it is included in future revisions for clarity.
>
> - Q6: Discussion about cutting planes for neural networks by acknowledging prior work on references
>
>    We acknowledge that our discussion on prior work was brief, and we appreciate the opportunity to rectify this by acknowledging significant contributions in this area.
>
>    1. We will cite and acknowledge Prior Works:
>
>       - Anderson et al. propose cutting planes specifically designed for neural networks by leveraging convex-hull constraints.
>       - [5] explores empirical bounds on the linear regions of deep rectifier networks, contributing to the understanding of the behavior and optimization of neural networks.
>       - [6] discusses partition-based formulations for optimizing ReLU neural networks, providing a framework for mixed-integer optimization in this context.
>       - [7] Section 4.2.4 offers a comprehensive overview of cutting planes and related techniques applied to neural networks, situating our work within a broader context.
>
>     2. Our work differs from Anderson et al. [1] in several key aspects:
>
>         - **Cutting Plane Design:** While Anderson et al. focus on selecting the most violated constraints among an exponential number of convex-hull constraints in **one layer**, our constraint can involve neurons from **any layer**, and our constraint can be easily found in infeasible subproblems during BaB.
>         - **Computational Expense:** Anderson et al. propose a linear-time method for constraint selection, which can still be computationally intensive for large-scale problems. In their experiments, they use neural networks within 2000 ReLUs, in contrast, we can scale to any network that beta-CROWN works, such as CIFAR100-ResNet-large, a network with 15.2M parameters and 286820 ReLU neurons.
>
> - Q7: Comment on Corrections to Writing
>
>     Thank you for your detailed feedback on the writing. We appreciate the opportunity to improve the clarity and precision of our paper. We will carefully review and revise the specified sections to incorporate these corrections.
>
> - Q8: What does "solve these cuts" mean?
>
>     To clarify, we don’t mean to generate the cut,  the term "solving cuts" was intended to describe the process of incorporating generated cuts—as additional constraints—into the linear programming (LP) problem. This process involves re-solving the LP relaxation with these new constraints to tighten the bounds of the problem.
>
>     In our revision, we will replace the phrase "solved these cuts" with "solving the LP relaxation with cuts included, which should more accurately reflect the process.

---

> > ### Comment · Reviewer_iWFh · 2024-08-12
> > **Follow up comments**
> >
> > I appreciate the effort of the authors with my questions. Here is a brief follow up on each point. I would appreciate a brief rebuttal on those.
> >
> > - Q1: This is exactly what most of the scholarship on mathematical optimization does: it tailors these general-purpose techniques to a particular problem at hand; yours is no different. By acknowledging that your contribution fits in this broader theme and adapts tried and proven methods (or rediscovers them, which is fair to say since we can assume you were not aware of those), you are doing less "marketing" and better scholarship. In my opinion, that would make your paper considerably better.
> >
> > - Q2: This is helpful for perspective. Please include those in the paper itself.
> >
> > - __Q3: This is wrong. Effective branch-and-bound works by defining disjunctions. Not only overlaps are ineffective, but in your case they lead to an algorithm that would never terminate (if you were really branching on $x$ that way, rather than on $z$ as you probably are). By branching on $x \leq 0$ and $x \geq 0$, the second subproblem is identical to the original problem (since $x \geq 0$ if $x$ is the output of a ReLU).__
> >
> > - Q4: Perfect!
> >
> > - Q5: Please add a note in the main paper linking to that.
> >
> > - Q6: Your comments about [5] and [6] do not cover cutting planes at all. Not worth including those references if you are not going to explain the cutting planes in those papers.
> >
> > - Q7: Good.
> >
> > - Q8: This is great. Please add some of this to the main paper.

---

> > > ### Author Response · Authors · 2024-08-13
> > > **We greatly appreciate your constructive feedback! Further clarifications on Q3**
> > >
> > > We greatly appreciate your timely response and found your questions and feebacks very insightful. To follow up on your questions, we want to clarify more, especially on **Q3**.
> > >
> > > Q1, Q2, Q5, Q6, Q8: Thank you for your valuable feedback. We will be sure to add/cite/rephrase our paper based on your suggestions here. They greatly helped us to improve our paper.
> > >
> > > Q3: We apologize again for the confusion, and we provided a more detailed answer here since there is no character limit any more. In the formulation of many prior papers (for example, beta-CROWN, Wang et al. NeurIPS 2021), $z$ was not explicitly included in the optimization formulation. However, when $x$ is branched to $x \leq 0$ and $x \geq 0$ cases, **the optimization problem is also changed**, essentially equivalent to branching $z$, as detailed below.
> > >
> > > For an unstable ReLU neuron, before branching, the ReLU function $y = ReLU(x)$ is relaxed using the "*triangle relaxation*":
> > >
> > > $y \geq 0$
> > > $y \geq x$
> > > $y \leq \frac{u}{u -l} (x - l)$
> > >
> > > Where $l$ and $u$ are the bounds of ReLU input ("preactivation bounds"). Note that **this formulation is without $z$**, but it is equivalent to the formulation with $z$ but with the relaxed variable $0 \leq z \leq 1$ projected out, forming a linear relaxation of ReLU. (note that our paper used $\hat{x}$, but I used $y$ to make the response easier to read)
> > >
> > > After branching, the neuron becomes a linear function in each branch. When $x \geq 0$, this ReLU neuron will be in the active region, and the **triangle relaxation is replaced by $y = x$**; when $x \leq 0$, the **triangle relaxation is replaced by $y = 0$** (following Beta-CROWN). So, the optimization formulation changed after branching on $x$. This is equivalent to branching on $z$: in the formulation where $z$ appears (Appendix A.1), when we set $z = 1$, removing redundant constraints will yield $y = x$; when we set $z = 0$, removing redundant constraints will yield $y = 0$.
> > >
> > > So when we say we are branching on $x$, the optimization formulations will change after branching - **it is *not* simply applying the constraints $x \geq 0$ or $x \leq 0$ directly on the triangle relaxation**; that would be incorrect as you pointed out. This branching procedure will be the same as branching $z$. In our paper, our cutting plane actually requires the formulation with $z$ since cuts were added to these variables. We will follow your suggestion to say we branch on $z$ instead of $x$ to make the setting more clear, and also add the above discussion to the appendix to avoid future confusion.
> > >
> > > Thank you again for your very constructive feedback. We hope all the questions have been addressed now, and we sincerely hope you can reevaluate our paper based on our response. Feel free to let us know if you have any additional questions for us.

---

> > > > ### Comment · Reviewer_iWFh · 2024-08-13
> > > > **Follow up 2**
> > > >
> > > > I appreciate the explanation regarding Q3. It would indeed be better to talking about branching on $z$ directly in the paper, but I would also recommend adding the comments about how this is done with the triangle relaxation in the appendix - with a brief mention in the paper to them.
> > > >
> > > > Regarding Q6, please make sure to revisit the references to explain them properly.

---

> ### Author Response · Authors · 2024-08-13
> **Feedback on Q3 and Q6: Enhancements and Clarifications**
>
> Thank you for your insightful comments and suggestions regarding Q3 and Q6. I've considered them and would like to share the following points:
>
> For Q3, we agree that it would be beneficial to discuss branching more directly in the main paper. We will add a clarification on how this is done with the triangle relaxation in the appendix, with a brief mention in the main text to direct readers there.
>
> Regarding Q6, we didn't go into depth in our initial response due to word limitations. However, here's the official version we are going to add to the related work section:
>
> [5] investigates the use of parity constraints as a cutting plane method in MILP for ReLU neural networks. These constraints are instrumental in defining a convex hull of feasible assignments, thereby improving the accuracy of approximating the number of linear regions and enhancing the network's expressiveness. While parity constraints (XOR cuts) significantly improve MILP performance by separating assignments with specific properties, they can also increase computational complexity due to their potential exponential growth with the number of variables. It is noteworthy that the XOR constraint used in [5] to construct the convex hull is similar to ours. However, their cut is constructed by solving the primal problem using the MILP solver, whereas ours is derived from infeasible domains in the inexpensive BaB.
>
> [6] delves into the use of cutting planes derived from convex hull constraints to optimize trained ReLU neural networks within MILP. These cutting planes serve as tightening constraints, effectively excluding infeasible solutions and improving solution quality. A notable feature of the proposed method is its linear-time approach to selecting the most violated constraints, which enhances optimization efficiency. By integrating these cutting planes into a partition-based formulation, the method achieves a balance between model size and tightness during optimization. However, the generation and integration of these cuts can be computationally expensive, and not all MILP solvers may support cut generation, potentially limiting their applicability in some scenarios.
>
> Please let us know if there are any further adjustments or if you'd like to discuss this in more detail.

---

> > ### Comment · Reviewer_iWFh · 2024-08-14
> > **Last comment**
> >
> > This is a good discussion of [6], which in a sense extends Anderson et al (to save you some space). The description of [5] is not correct, but this is a lesser important reference, so don’t worry about it.

---

### Official Review · Reviewer_1a6g · 2024-07-09

**Soundness:** 2
**Presentation:** 1
**Contribution:** 2
**Rating:** 6
**Confidence:** 5

**Summary:**

The paper presents BICCOS: a method to derive cutting planes for use within a state-of-the-art neural network verification framework based on branch and bound (BaB). Given verified (UNSAT) subproblems, BICCOS tries to find a subset of the employed branching choices that led to the verification result, and applies a cutting plane that prunes this subtree from the rest of the BaB procedure. Engineering improvements (going over multiple BaB tree and branching choices in parallel as a pre-processing step) are also presented. The experimental results suggest moderate improvements upon the state-of-the-art over the considered benchmarks.

**Strengths:**

The idea behind BICCOS is fairly simple, yet relatively novel in the context of neural network verification. Given the additional overhead linked to the "strengthening" procedure (recomputing bounds after removing branching decisions), which is required for the overall algorithm, one may think that the overall approach may not pay off. The experiments show that it does, although somewhat marginally, I would believe.

**Weaknesses:**

**Presentation.**
The paper feels quite rushed, and the quality of the presentation definitely needs to improve to meet the NeurIPS bar.
The figures are fairly small (especially Figure 2) and fairly hard to read on paper. I would suggest that the authors remove the shadows too, which make things harder. The Tables are also fairly hard to read on paper. The text still has some typos (e.g., "BICwhere" in line 139). The example from Figure 1 does not correspond to the text in lines 160-166 (x_3 >= 0 vs x <= 0) or to Figure 2a. In lines 308-310 the text suggests that the comprehensive BICCOS configuration performs the best in all cases: this is not what appears from Table 3.

**Feasibility.**
This is linked to the presentation but it's important enough to stand as a separate point. Page 4 repeatedly speaks of infeasibility in a context where I think it's technically incorrect. I think that the fact that a subproblem lower bound is positive does not imply infeasibility: it could very well be that both its lower and upper bounds are positive (with UB > LB), simply suggesting that the minimum for that subproblem (but of course not necessarily for the original problem) is positive. This means that the counter-example search is infeasible, but not the variable assignment (the series of branching decisions). In order to prove infeasibility of the subproblem, one would need to either show that the local UB is smaller than the subproblem LB, or show that the subproblem LB would go to infinity in the limit for iterations (that is, the underlying dual problem is unbounded). All the arguments being made still apply even for feasible yet verified subproblems: the goal is simply to exclude BaB subregions which we know already will lead to positive lower bounds (hence pruning the tree). But this terminology should be adapted to avoid any confusions.

**Results.**
While the fact that the proposed approach works is interesting (see *strengths* above), I do not think the presented experimental results are particularly impressive. Most of the improvements over GCP-CROWN (or Beta-CROWN, when GCP-CROWN can't be applied because of scalability issues) are fairly small. Furthermore, more granularity in the results would be needed (see questions). While this should not be a problem for acceptance on its own, I think it is when combined with the presentation issues above.

--------

**Post-discussion.** I am increasing my score to 6 following the discussion with the authors. I encourage them to acknowledge the shortcomings of the proposed approach in the next version, and to improve the presentation as discussed.

**Questions:**

1) It is repeatedly claimed that the procedure is specific to NN verification, but I think such an approach would apply more generally to any BaB procedure (finding subsets of branching decisions that led to a positive verification result and using that as cutting plane), or at least on any BaB procedure for MILPs. Could you please elaborate on this?
2) As commonly done in previous work (for instance, Beta-CROWN), plots showing the number of verified properties within a given runtime are needed to fully assess the trade-offs associated to the proposed approach. For instance, how much does it slow verification down on easier properties?
3) It seems to me that Table 2 reports the best configuration across those in Table 3, for each BICCOS row. Could the authors clarify this?

**Limitations:**

Limitations are appropriately addressed in the conclusions.

---

> ### Author Rebuttal · Authors · 2024-08-07
>
> We thank the reviewer for the constructive comments and valuable questions. We want to clarify a few key misunderstandings about feasibility and results. We hope the reviewer can reevaluate our paper based on our response below:
>
> - For Weakness
>
>     * Presentation.
>
>         For presentation, we will fix typos and adjust format according to the reviewer's feedback and  revise the text to accurately reflect the results shown in Table 3, clarifying that the comprehensive BICCOS configuration does not perform best in all cases.
>
>     * Explanation of feasibility.
>
>         We realize the potential for confusion. Our formulation is in line with your last statement: We are only interested in the regions that may potentially contain adversarial examples, i.e. there exist inputs $x$ such that $f(x) <= 0$. Therefore, if the lower bound does become positive, the existence of adversarial examples in this subdomain can be excluded (infeasible).
>         Generally, in neural network verification, the safety property (non-existance of adversarial examples) is negated and posed as a satisfyability problem. Tools then report SAT if adversarial examples exist and UNSAT if the safety property holds. This implies that the input $x$ is restricted not only to the given input area, but also to those $x$ where $f(x) <= 0$. If no adversarial example exist, no $x$ with $f(x) <= 0$ exists, so the assignment becomes infeasible. Often, this constraint is only used indirectly: First, a lower bound of $f(x)$ is computed. Then, if it is positive, this implies that the underlying assignment is in fact infeasible, as the constraint $f(x) <= 0$ would always be violated. There is also work [1,2] on directly incorporating this constraint into the optimization process.
>         While the implicit conversion from “lower bound > 0” to “infeasible” is common in the neural network verification community, we recognize the need to make this step explicit. We will rewrite the respective sentences accordingly to avoid confusion, but we want to emphasize that the our existing theoretical results are sound and not affected by these changes.
>
>         [1] Kotha, S., Brix, C., Kolter, J. Z., Dvijotham, K., & Zhang, H. (2023). Provably bounding neural network preimages. Neurips 2023.
>
>         [2] Pengfei Yang, Renjue Li, Jianlin Li, Cheng-Chao Huang, Jingyi Wang, Jun Sun, Bai Xue, and Lijun Zhang. Improving neural network verification through spurious region guided refinement. Tools and Algorithms for the Construction and Analysis of Systems, 2021b.
>
> - Q1. Can BICCOS be applied to any BaB procedure?
>
>     BICCOS is specialized to the branch-and-bound procedure of most SOTA neural network verification tools. While neural network verification constitutes a sub-problem of the more general MILP problem set, the respective tools have been tuned to its specific kind of problems.
>
>     Specifically, the ReLU activation functions in neural networks are difficult for regular MILP solvers to handle, as they require extensive branching to cover all possible combinations of assignments. Neural network verification tools have developed specialized techniques to deal with those non-linear activation functions by overapproximating them and - crucially - improving this overapproximation iteratively using GPU-acceleration.
>
>     However, by tuning the neural network verification tools toward this specific subset of tasks, they have become non-ideal (or impossible) to apply to regular general verification problems. Therefore, BICCOS cannot be directly applied to generic MIP problems not relavent to non-neural network verification.
>
> - Q2. How much does BICCOS slow down the verification on easier properties?
>
>     Thank you for pointing out the need for plots to demonstrate the trade-offs of our approach, in response to your query, **Fig.2 in the pdf** file illustrates the number of verified properties within various runtime thresholds. This visualization helps to clarify the performance trade-offs associated with our approach. As indicated in the figure, the slowdown experienced with our method is relatively minor for simpler properties, which are often verified before the implementation of cuts. For more complex instances, however, the benefits of our approach become more evident, with a notable improvement in time efficiency for verification. This trend suggests that while our method introduces a slight delay in simpler cases, it significantly enhances performance on more challenging properties, providing a net gain in efficiency across a diverse set of scenarios. We believe that this balanced approach is beneficial for practical applications where varying levels of difficulty are encountered.
>
> - Q3 Does Table 2 report the best configuration?
>
>     Thank you for your observation. Table 2 indeed reports the best configuration for each BICCOS row as identified among those listed in Table 3. This approach ensures that the reported settings are optimal for each benchmark. For instance, large models perform best with MTS, while small models benefit from MIP cuts, etc., on each dataset. This methodology is consistent with common practices in the field. E.g. in the VNN-COMP, teams often fine-tune their tools for each benchmark set. Crucially, we note that we did not explore a large set of hyperparameters, and using all BICCOS features (MILP cuts, constraint strengthening, multi-tree search) is best for all but 2 benchmarks, where it is outperformed by the tuned version (multi-tree search disabled to avoid its overhead) by only 0.5 percentage points.
>
> We thank you again for the valuable comemnts and we hope our weaknesses (especially on infeasibility) has been addressed. We hope you can reevaluate our paper based on our response. Thank you.

---

> > ### Comment · Reviewer_1a6g · 2024-08-11
> > **Thank you for your response**
> >
> > I thank the authors for their response. I appreciate the willingness to improve the presentation of their work in a future version, and thank the authors for the clarification on their use of feasibility.
> >
> > Unfortunately, I am still leaning towards rejection, as I still believe the experimental improvements to be somewhat marginal, and I still think that, presentation-wise, the submission feels a bit rushed.
> >
> > I understand that it is common practice in VNN-COMP to tune and engineer a framework to a given setting, but in my own view the goal of a paper is slightly different. The shortcomings of the presented approach should have featured more prominently, along with a comprehensive explanation as to why something would not pay off in a given setup. Instead, the submission appears to be seeking to sweep the shortcomings of the full configuration under the rug. For instance, line 309 even states "The comprehensive BICCOS configuration, incorporating all optimizations, achieves the highest verified accuracies across models.", which is either incorrect or misleading.
> > Related to this, I think the paper should have included a more prominent description of the overhead of the framework on easier properties. I appreciate the inclusion of Figure 2 in the response, albeit I think it is incomplete and would have read better in a log scale over time (as it is, the overhead on slow properties is hard to quantify as a share of runtime). An updated Figure 2, also featuring other baselines such as GCP-CROWN, which is expected to perform much better than Beta-CROWN on harder properties, should definitely appear in the next version of the work.

---

> > > ### Author Response · Authors · 2024-08-13
> > > **Discussions on our presentations and results (part 1/2)**
> > >
> > > We are very grateful for your timely response. Following your constructive advice, we would like to clarify a bit more about the presentation of our paper and the significance of our results.
> > >
> > > > the experimental improvements to be somewhat marginal
> > >
> > > We want to point out that the room for improvement for many benchmarks is not big—the **verification lower bound is quite comparable to the PGD upper bound**, so a massive improvement cannot be shown if we directly read these numbers. For example, in Table 2, we have MNIST CNN-A-Adv (74.0% vs 76.5%), CIFAR CNN-A-Adv (49% vs 50%), CIFAR CNN-A-Adv-4 (48.5% vs 49.5%), and CIFAR CNN-A-Mix-4 (56.5% vs 57.5%). Although these standardized benchmarks have been widely used in the literature, they have only a few percentage points left for improvement.
> > >
> > > In fact, if you look at the gap between the lower and upper bound, we did get a quite pronounced improvement. For example, for MNIST CNN-A-Adv, GCP-CROWN has a **4.5%** gap between lower and upper bound, but we have only **2.5%** gap. That is a ~44% improvement on reducing this gap. Also, in Table 1, on oval21, we completely **close the gap** between lower and upper bound; on cifar10-resnet, the number of unsolved instances (gap) is **reduced from 9 to 6** compared to GCP-CROWN; on cifar100-tinyimagenet, the gap is **reduced from 25 to 18**.
> > >
> > > **The verification community has been working hard to close this gap** (see [ref. A] below, Intro section), and in fact, the few instances remaining in each benchmark reported here, are all very challenging ones. For example, in GCP-CROWN paper, which completely solved the oval20 benchmark (their Table 1), only improves the verified instances from 98% to 100% (CIFAR-10 Wide) and 97% to 100% (CIFAR-10 Base). Number-wise, it is just a "marginal" (a few percentage points) improvement similar to the improvements we report, **but it is actually quite a big achievement since no algorithm could solve these remaining hard instances**. Similarly, in [ref. A], which aims to improve the upper bound to close the gap, they also only demonstrated improvements on very few hard instances - **their improvement is less than 0.5%** if evaluated on the entire dataset as we did.
> > >
> > > [ref. A] A Branch and Bound Framework for Stronger Adversarial Attacks of ReLU Networks, Zhang et al., ICML 2022
> > >
> > > > the submission appears to be seeking to sweep the shortcomings of the full configuration under the rug.
> > >
> > > > line 309 even states "The comprehensive BICCOS configuration, incorporating all optimizations, achieves the highest verified accuracies across models."
> > >
> > > Following your suggestions, we will rephrase the sentence as "Our results show that the addition of BICCOS cuts is overall beneficial. When a MIP solver is feasible, BICCOS can be combined with the MIP cuts in GCP-CROWN to potentially further improve the verified accuracy of GCP-CROWN, demonstrating the quality and effectiveness of our cuts. When MTS is used, it may further improve the verified accuracy on some benchmarks, but the overhead of MTS may reduce the verified accuracy on some benchmarks."
> > >
> > > We will also replace the numbers under the BICCOS column in Table 2 with the numbers with MTS and MIP cuts enabled (if an MIP solver can scale to this setting), even in the case where MTS slows down verification due to overhead. In fact, we have optimized our software further to reduce the overhead of MTS, and the potential overhead has now become smaller. Eventually, we will enable all BICCOS components to be the default in our to-be-released verifier. We will produce a message to users when MTS introduces too much overhead compared to the overall timeout threshold and suggest users turn off this feature.

---

> > > > ### Author Response · Authors · 2024-08-13
> > > > **Discussions on our presentations and results (part 2/2)**
> > > >
> > > > (continued, part 2/2)
> > > >
> > > > > The shortcomings of the presented approach should have featured more prominently, along with a comprehensive explanation as to why something would not pay off in a given setup.
> > > > > description of the overhead of the framework on easier properties
> > > >
> > > > **We will add this paragraph, "Shortcomings and Overhead,"** to the experiment section of our paper:
> > > > We note that the multi-tree search (MTS) and cut strengthening procedure incur additional overheads compared to the baseline verifiers. The MTS has about 2 - 10 seconds overhead in the benchmarks we evaluated, which may increase the verification time of very easy instances that can be sequentially immediately verified in branch and bound. The BICCOS cut strengthening is called during every branch-and-bound iteration, but its cost is *accumulatively* about 3 - 20 seconds for hard instances in each benchmark, which run a few hundred branch-and-bound iterations. We note that typically, the performance of the verifier is gauged by the number of verified instances within a fixed timeout threshold, thus when the threshold is too low, the added cutting planes may not have time to show their performance gains due to the lack of sufficient time for the branch-and-bound procedure.
> > > >
> > > > We note that such a shortcoming is also shared with other verifiers which require additional steps for bound tightening; for example, the GCP-CROWN verifiers have the extra overhead of a few seconds of calling and retrieving results from the MIP solver, even when the MIP solver is running in parallel with the verifier. The final verified accuracy can justify whether the overhead is worth paying.
> > > >
> > > > > An updated Figure 2, also featuring other baselines such as GCP-CROWN, which is expected to perform much better than Beta-CROWN on harder properties, should definitely appear in the next version of the work
> > > >
> > > > > (the figure) would have read better in a log scale over time
> > > >
> > > > We have produced these figures with all baselines in log scale locally, but unfortunately we cannot update the figure on the rebuttal PDF anymore and NeurIPS disallows the use of external links. We will be sure to include these in our final revision. To describe these figures verbally, for easy instances (solved within a few seconds), Beta-CROWN is the fastest. GCP-CROWN has the overhead of initializing MIP solver and our method has the overhead of MTS, so they are slower on easy instances; however, on hard instances which distinguish the power of verifiers, BICCOS shows clear benefits (the BICCOS line is bent towards the lower right corner) on all benchmarks. With a log-scale, it is clear to show the overhead on the easy instances and the benefits on hard instances.
> > > >
> > > > Thanks again for your valuable feedback. We want to emphasize that there is no technical or soundness concern in our paper, and the **presentation issues will be fixed in our final version, as we have detailed in our response**, especially on the presentations of the tables/figures and the discussions of the shortcomings and overhead. We sincerely hope you can reevaluate our paper based on our technical contribution, and we hope that these easy-to-fix problems will not become a roadblock to accepting our paper.

---

### Official Review · Reviewer_vA2E · 2024-07-12

**Soundness:** 3
**Presentation:** 2
**Contribution:** 2
**Rating:** 4
**Confidence:** 4

**Summary:**

The paper extends GCP-CROWN, an existing toolkit for the verification of neural networks which is based on GPU-accelerated bound propagation combined with a branch-and-bound (BaB) approach. The strength of the existing algorithm is its ability to incorporate cutting planes into the bound propagation process. GCP-CROWN uses cutting planes generated by a Mixed Integer Linear Programming (MILP) solver which is run in parallel to the bound propagation, however, MILP solvers generally do not scale to large problems and only generate generic, problem-independent cutting planes.
The authors propose a new approach to generate cutting planes called BICCOS which works by exploiting information from verified branches in the BaB tree. Once a branch is verified, the idea is to remove a number of constraints from the branch in an attempt to obtain a subset of constraints that is sufficient for obtaining a "verified" result. If successful, a cut which is valid for all other branches in the BaB tree can be generated based on these constraints. The generation of cuts is run during the normal verification process, but the authors propose adding a presolve step that initially generates multiple shallow BaB trees in an attempt to create a pool of cuts before starting the standard BaB phase from GCP-CROWN.
The experimental evaluation shows that BICCOS scales well, is able to generate cuts for problems that the MILP solver employed in GCP-CROWN can't scale to, and outperforms many competing tools.

**Strengths:**

- Neural Network Verification is a relevant research topic
- A cut generation method that scales to larger networks as well as cuts that are problem-specific and less generic than those generated by a MILP solver are useful contributions.
- The method outperforms most other toolkits in the experimental evaluation

**Weaknesses:**

- The work is somewhat incremental compared to GCP-CROWN
- When comparing the performance of GCP-CROWN and BICCOS (base) in Table 3, this seems to indicate that the newly introduced cuts are weaker than the MILP cuts. Including presolve (BICCOS(with multi-tree)) improves performance compared to GCP-CROWN in only some instances. It's good to have a cut generation method which scales to larger networks, but the method would be a lot stronger if the BICCOS cuts alone outperformed the generic MILP cuts.
- A comparison with existing cuts, such as the ones from [1], is missing. In the related work section the authors only state that Venus (which implements the cuts from [1]) delivers weaker empirical results than their approach. However, the contribution of this paper are the new cuts and comparing the BICCOS cuts in a GPU-enabled bound propagation framework to the cuts by Botoeva et al. implemented in a MILP-based verifier (which can't make use of GPU acceleration) is not fair since it is well-known in the literature that bound propagation frameworks outperform MILP verifiers. To assess whether the BICCOS cuts are more effective than the cuts in [1] they should be implemented in the same general framework. Without this comparison it is hard to judge the contribution of this work.
- Appendix A2: This part of the appendix is either very unclear or has a lot of typos. There are a lot of expressions like $\sum_{i \in \mathcal{Z}^+} z_i + \sum_{j \in \mathcal{Z}^-} (1 - z_i)$. Do the authors mean to write $\sum_{i \in \mathcal{Z}^+} \left ( z_i + \sum_{j \in \mathcal{Z}^-} (1 - z_i) \right )$? If so, the extra set of brackets should be added. If the authors actually do mean to write $\sum_{i \in \mathcal{Z}^+} \left ( z_i \right ) + \sum_{j \in \mathcal{Z}^-} (1 - z_i)$ then the $z_i$ in the second sum makes no sense, should this be $z_j$ then? This unclarity/mistake appears in the two equations between line 516 and 517 (which aren't labeled so I can't refer to them), in line 518, in the first equation between line 518 and line 519 and the second equation between line 518 and 519 (these also aren't labeled so I can't refer to them directly).

### Minor points
- Line 10-11: cutting planes constraints --> cutting **plane** constraints
- Line 60: proposed --> **propose**
- Line 67: discussed --> **discuss**
- Line 98: The paper states "If $f^* > 0$, it is unclear whether the property might hold" --> Shouldn't this be "If $f^* < 0$" (i.e. the inequality being flipped?)
- Line 98: proerty --> pro**p**erty
- Line 109: Remove "the" (sentence should be "most existing NN verifiers use cheaper methods such as (...)")
- Line 139: Remove **BIC** at the beginning of the line
- Line 143: they --> **the authors**
- Figure 2a): these subproblems already includes the constraint --> these subproblems already **include** the constraint
- Line 194: along --> **alone**
- Line 198-199: using as fewer variables as possible --> using as **few** variables as possible
- Line 202: we performs a re-verification step, where it recomputes the lower bound --> we **perform** a re-verification step **which** recomputes the lower bound
- Line 208: we propose --> **W**e propose
- Table 3: For MNIST, CNN-A-Adv the "Ver%" for BICCOS (with MIP cuts) is $0.71$. Is this a typo, what is the correct number?
- Appendix A.2: The authors derive two equations by "taking a negation of this equation". I find this part a bit unclear, do they mean that if the equation holds then this is equivalent to one of the two new equations holding? Or do both of the two new equations need to hold?
- Line 547-549 in Appendix A4: The authors write "on the CIFAR CNN-B-Adv model, BICCOS with multi-tree search explores $2.54 \times 10^3$ branches, significantly lower than the $1.57 \times 10^3$ branches explored by the base BICCOS version." However, as far as my understanding goes, $2.54 \times 10^3$ is not a **smaller** but a **larger** number than $1.57 \times 10^3$ so the sentence here makes no sense. I tried to double-check this but the same numbers are reported in the table below (I assume that "BICCOS with multi-tree search" in the text is the same as "BICCOS (with Presolve)" in the table). Could the authors clarify what their point is here?

### References
[1] Botoeva, E., Kouvaros, P., Kronqvist, J., Lomuscio, A. & Misener, R. (2020) Efficient Verification of ReLU-Based Neural Networks via Dependency Analysis. In: Proceedings of the AAAI Conference on Artificial Intelligence. 3 April 2020 pp. 3291–3299. doi:10.1609/aaai.v34i04.5729.

**Questions:**

- Table 3: Why does BICCOS with multi-tree perform worse than BICCOS (base) for CNN-A-Adv-4, do the authors have an explanation for this?
- Why is BICCOS run with a shorter timeout compared to other algorithms? Also why is MN-BaB run with a 600s timeout and then $\beta$-CROWN, GCP-CROWN and BICCOS are run with a 200s timeout in Table 2, but then in Table 3 the footnote seems to suggest that BICCOS is run with a 200s timeout while $\beta$-CROWN and GCP-CROWN use a longer timeout? This seems inconsistent. The experiments would be more informative if all algorithms were run with the same time budget as is usual practice e.g. in VNNComp.
- In Table 1/2 does BICCOS use cuts from a MILP solver (if the solver scales to the problem) or only the newly introduced cuts?
- Table 3: For CNN-A-Mix-4 the BICCOS-MIP approach has a verified accuracy of 56.5% but BICCOS-all has 56%. What is the authors' intuition here regarding why adding the multi-tree approach worsens performance, do they think this is an issue/can be avoided?
- Line 302-310: Could the authors clarify what each variant of the algorithm includes here? The text makes it sound a bit like the authors start from BICCOS (base) and then gradually add other components, but does BICCOS (with multi-tree) also include MIP cuts? If so, the performance drop from 52 to 51.5% on e.g. CIFAR CNN-B-Adv would be surprising, any explanations?

**Limitations:**

Limitations are sufficiently addressed.

---

> ### Author Rebuttal · Authors · 2024-08-07
>
> We thank the reviewer for the constructive comments and valuable questions. We hope the reviewer can reevaluate our paper based on our response below:
>
> * For Weakness.
>
>     - W1. The work is somewhat incremental compared to GCP-CROWN
>
>         GCP-CROWN and our work make orthogonal contributions. GCP-CROWN does not provide an efficient way to find cuts as we do.  GCP-CROWN primarily focuses on solving cuts provided to it without engaging in the cut-finding process. We just use GCP-CROWN as a solver to do bound propagation with constraints. On the other hand, our work introduces an efficient algorithm specifically designed to find these cuts. This distinction highlights a significant contribution of our method: the ability to identify and generate cuts, not just solve them. By developing an algorithm that efficiently finds cuts, we add a valuable tool to the existing framework, enhancing the overall effectiveness of neural network verification.
>
>     - W2. Performance Comparison of MILP cuts and BICCOS.
>
>         We agree that MILP solver cuts are inherently powerful due to the solver's comprehensive approach from a lot of previous works [1] during the past decades. However, a critical limitation of MILP solver cuts is their scalability. As the network size increases, i.e., above 4M parameters, the complexity and computational resources required for MILP solvers to generate cuts even initialize the model become prohibitively high. In our experiment, we found MILP could not scale to large networks like cifar100 (5.4-31.6 M) and tiny-imagenet (14.4 M).  This limits their practical applicability to larger neural networks.Our BICCOS method, while producing slightly weaker cuts compared to MILP, offers significant advantages in terms of scalability.
>
>         [1] Wolsey, L. A., & Nemhauser, G. L. (2014). Integer and combinatorial optimization. John Wiley & Sons.
>
>     - W3. Comparison with Venus2
>
>         **Fig. 1 in the PDF file** shows a comparison with Venus2. We emphasize on making a fair comparison on the strengths of cuts. Since Venus2 uses an MILP solver to process its cuts, in these experiments we do not use the efficient GCP-CROWN solver. Instead, we also use an MILP solver to handle the BICCOS cuts we found. This ensures that the speedup we achieve is not coming from the GPU-accelerated GCP-CROWN solver. Since our cut generation relies on the process with BaB, we first run BICCOS to get the cuts, and then import the cuts into the MILP solver.
>
>         We note that Venus uses branching over ReLU activations to create multiple strengthened MILP problems. On the other hand, we only create one single MILP and do not perform additional branching. Therefore, our MILP formulation is weaker. The fact that we can outperform Venus2 anyway underlines the strength of the generated cuts by BICCOS.
>
>     - W4. Appendix A2 function typo.
>
>         A We apologize for the confusion caused by the typo. We intended to write         $$ \sum_{i \in \mathcal{Z}^{+}} z_i + \sum_{i \in \mathcal{Z}^{-}} (1 - z_i) \leq |\mathcal{Z}^+| + |\mathcal{Z}^-| - 1 $$
>
> * For minor issues
>
>     - Typos. Thank you for the detailed review, we will fix the typos. The correct number is 71, not 0.71.
>
>     - Explanation in Appendix A.2. When we refer to "taking a negation of this equation," we mean that if the original equation holds, it leads us to the two new conditions. However, given the context and the bounds on $z$, only the first equation needs to hold.
>
>     - Explanation in Appendix A4. The sentence was meant to highlight the efficiency of the multi-tree search despite exploring a larger number of branches because **the multi-tree search domains are also counted in the total number of branches**. The multi-tree search method inherently examines multiple trees, thereby increasing the number of branches explored. These domains represent different subproblems that are explored simultaneously. Although this increases the branch count, the exploration within each domain is optimized, leading to faster convergence and overall reduced computation time. We will correct the text to reflect this explanation accurately. Thank you for your careful review and for helping us improve the clarity of our paper.
>
> * For questions
>
>     - Q1. Why does BICCOS with multi-tree perform worse than BICCOS (base) for CNN-A-Adv-4?
>
>         For the CNN-A-Adv-4 dataset, BICCOS with multi-tree search performs worse than BICCOS (base) primarily due to the additional time cost associated with the multi-tree search approach. This dataset contains 200 instances, and each instance has 10 predicted classifications, requiring us to validate 9 properties so in the worst case we have to perform a multi-tree search for each property.
>
>     - Q2. Timeout difference in comparisons
>
>         We apologize for the confusion. The MN-BaB results were copied from the VNN-COMP 2022 report. We did not reproduce those ourselves, though we use the same hardware for our experiments as they did. In table 3, BICCOS, beta-CROWN and GCP-CROWN all have a timeout of 200s. We will update the table caption accordingly. The increased timeout for MN-BaB may increase the percentage of verified instances. However, we can still outperform it.
>
>     - Q3. Question in Table 1 \& 2.
>
>         In Table 1/2,BICCOS uses cuts from BICCOS base + multi-tree search + MIP cuts from CPLEX
>
>     - Q4. Question in Table 3.
>
>         In table 3, the performance degradation by multi-tree search is caused by the associated computational overhead.
>
>     - Q5. Clarification what each variant of the BICCOS uses
>
>         - Biccos base: only contains the cut inference during regular bab,
>         - BICCOS (with multi-tree): includes BICCOS base but not MIP cuts,
>         - BICCOS (all): includes base multi-tree and MIP cuts.

---

> ### Comment · Reviewer_vA2E · 2024-08-08
>
> Thank you very much to the authors for the clarification of the points that I raised and for answering my questions. I appreciate the thorough response regarding concerns W1/W2 and the explanations regarding what the main contributions are from your side.
> The additional comparisons as a response to W3 are very useful, thank you for this. I think it would be helpful if this was included in the appendix of the paper.

---

> > ### Author Response · Authors · 2024-08-10
> > **We thank the reviewers again and please let us know if you have any further questions before the discussion is closed**
> >
> > Thank you for your constructive feedback and for acknowledging our responses. We're glad that the additional comparisons addressing W3 were helpful, and we agree that including them in the appendix would be beneficial. We will make sure to incorporate this in the final version of the paper. We hope these updates might lead you to reconsider your score. Your insights are much appreciated and please let us know if you have any further questions before the discussion is closed
> >
> > Best Regards,
> >
> > Anonymous Authors

---

### Official Review · Reviewer_SS1K · 2024-07-13

**Soundness:** 4
**Presentation:** 4
**Contribution:** 3
**Rating:** 7
**Confidence:** 4

**Summary:**

This work proposes a new approach to produce cutting planes in the context of branch-and-bound-based solvers for neural network verification. Whenever an infeasible subproblem is encountered in branch-and-bound, this method generates a cut from the conflicting assignment that led to the infeasible subproblem (initially redundant w.r.t. the remainder of the tree), and attempts to strengthen this cut by heuristically dropping some of the assignments and rechecking for infeasibility via the lower bound. This is further enhanced by using several parallel shallow trees to produce stronger cuts. This method is implemented on top of the $\alpha,\beta$-CROWN framework and provides meaningful computational improvements compared to various baselines on a set of benchmarks.

**Strengths:**

This paper provides a solid contribution to the area of neural network verification methods. It builds on top of cut-based branch-and-bound verification solvers by presenting a method to quickly infer cuts, which appear to be novel and computationally useful. In particular, they nicely leverage the fact that there are fast methods to produce bounds in NN verification, allowing us to quickly recheck cut validity. This makes for a clean and simple method, which has the advantage of not being too complicated to integrate with an existing cut-based BaB verification solver.

Both the set of benchmarks and baselines are reasonably extensive, and we can observe improvements in verifiability that are sufficient for a meaningful computational contribution, especially in the CIFAR instances in both VNN-COMP and SDP-FO, without much additional cost in computational time.  The paper is overall clearly written and the figures and algorithms are helpful.

**Weaknesses:**

In some of the benchmarks, the computational results may be somewhat incremental, but overall they are positive. There is potential room for improvement in parts of the methods (see Questions section below); in particular, it is not clear if the authors have explored variations of their constraint strengthening approach. In general, I do not see major weaknesses in this paper, though minor concerns are expanded on below.

**Questions:**

General comments:

1. I see that your variable-to-drop selection heuristic is based on their improvements to the lower bound in the tree. While this seems reasonable as a fast heuristic since you already have all the data, these improvements are not independent from each other since they are constrained over previous assignments, and thus there is some bias depending on the depth (i.e. if the assignments were done in a different order in the tree, you'd select different variables, but the constraint is the same). This makes me wonder if there is a better heuristic. Have you considered other approaches?

2. It seems that you try to continue strengthening the cuts based on a fixed drop percentage. I am curious if you have tried something more like a binary search approach over the verification bound? You can also add some sort of tolerance on the bound to stop searching when you are close enough to zero.

3. In Algorithm 1, lines 12-13, you add the cut, and then try to strengthen it again. It sounds like you could just add the best cut here, instead of adding all cuts throughout strengthening, since the best cut dominates the other ones.

4. I appreciate the explanation of the differences between this method and [7] and DPLL/CDCL in the Related Work section, as both of these were in my mind as I was reading the paper. However, I'd like to comment that the reasoning for CDCL makes it sounds like it is impossible in practice to use learned clauses from CDCL as cuts; rather, I believe this is more of a challenging engineering task than not being practically viable. Much like branch-and-bound has been customized for NN verification in a more effective way than generic MIP solvers, I do not see a reason why one would not be able to customize ideas from CDCL for your cut generation procedure. The learned conflicts can be naturally translated to the type of cuts you have in Sec. 3.1 of this paper (except already stronger) and then further strengthened in the same way. Given that CDCL is a tried-and-true method to produce good conflicts, I suspect that this might lead to better cuts and may be interesting future work that would already fit very well with what you have so far.

5. I would have liked to see some data to better understand the cuts generated. In particular, what is the fraction of infeasible nodes from which you were able to produce a (non-redundant) cut, and what was the total number of cuts? What was the average number of assignments that you were able to drop? This sort of data would reveal more information on the overhead of these cuts and how easy or hard it is to find them. I wonder if some cut selection procedure would make sense here, but I do not know if you have many or few cuts.

Comments on text:

6. In the MIP section, it would be useful to mention that MIP is also based on branch-and-bound, and modify the last sentence to include why a custom branch-and-bound is more effective than MIP branch-and-bound in practice (e.g. because MIP is based on solving LPs which can be expensive, etc.).

7. In Sec 3.1., I suggest preparing the reader to the fact that the cuts from Sec. 3.1 are redundant w.r.t. the tree until strengthened, instead of waiting until Sec. 3.2 to mention that. This is a question one would naturally have while reading Sec. 3.1, and it would make the reading easier if they already know the answer to that question.

8. Figure 2 is a bit too small to read especially when printed out. If possible, please make it larger.

9. Can you include in the text that your bound computation when strengthening a cut includes the global set of cuts? I see it in Algorithm 1, but I didn't see it in the text.

10. Could you include exactly how the trees differ from each other in the multi-tree approach? The text mentions that they explore a different set of branching decisions, but not how.

11. Could you expand on which methods/baselines use GPUs, and how they are used? This is important for a proper comparison.

12. Typos: Remove "BIC" in line 139, capitalize "we" in line 208, "instances" and double period in caption of Table 4.

**Limitations:**

The Limitations section is reasonable, covering cases where this approach does not work well.

---

> ### Author Rebuttal · Authors · 2024-08-07
>
> We thank you very much for your constructive feedback and for correctly recognizing our key contributions. We appreciate your support and very helpful feedback. We provided additional experiments as requested and clarified the key questions below:
>
> * Q1: Do You Consider Other Drop Heuristics?
>
>     We acknowledge that while our current approach, based on improvements to the lower bound in the tree, is efficient given the available data, it does introduce dependency biases due to the constrained nature of previous assignments. There might exist better heuristics that can be explored in future work, but our work is the first of this kind and we want to start with a simple and effective heuristic.
>
>     We have considered and tested a random drop heuristic and KFSB score heuristic as alternatives. Our benchmark results across SDP and oval22 indicated no significant improvement in performance compared to beta-CROWN, please refer to the following table. These results suggest two heuristics do not provide a robust alternative in this context.
>
>     | Dataset | Beta CROWN | Random Drop | KFSB Score | Influence Score |
>     |------------------|------------|-------------|------------|-----------------|
>     | cifar_cnn_a_adv | 44.50% | 44.50% | 44.50% | 47% |
>     | cifar_cnn_a_mix | 41.50% | 41.50% | 41.50% | 45.5% |
>     | cifar_cnn_b_adv | 46.50% | 46.50% | 46.50% | 49% |
>
> * Q2: Exploring Binary Search Approaches for Cut Strengthening
>
>     It’s correct that our current method employs a fixed drop percentage. We did not investigate a binary-search-based approach, as this would increase the number of verification queries. Instead, we recursively tighten (line 205) the cut further should the first query succeed. If it fails, re-introducing constraints would reduce the benefit of BICCOS, while increasing the associated overhead. However, we do acknowledge that this could be tuned further in future research. We did explore dropping only 30% of the constraints at a time, with no immediate benefit. This demonstrates that we are not sensitive to this hyperparameter.
>
> * Q3: Optimizing Cut Addition in Algorithm 1 by Selecting Only the Best Cut
>
>     We agree the strengthened cuts make the previous cuts obsolete. We will replace lines 12 and 13 of the algorithm with
>
>     ```
>     recursively_strengthened_cuts = constraint_strengthening(f, C_{new}, C_{cut}, drop_percentage)
>     if strengthening_was_successfull:
>         C_{cut} <- C_{cut} \cup recursively_strengthened_cuts
>     else:
>         C_{cut} <- C_{cut} \cup strengthened_cuts
>     ```
>
>     Intuitively, if the next rounds of constraint strengthening produce a better cut, we use these better cuts rather than the currently inferred cuts.
>
> * Q4: Integrating CDCL Learned Clauses as Cuts for Enhanced Cut Generation
>
>     We agree that CDCL can be used to generate cuts even though this will be a challenging engineering task. This is an interesting future work and we will rephrase our paper accordingly. Our current approach focuses on generating cuts and then strengthening them, which has proven to be both easy to implement and effective. While CDCL could enhance the initial cut generation, it does not inherently provide a mechanism to strengthen these cuts using the solver, a novel step in our paper that is crucial for performance.
>
> * Q5: Data Analysis on Cut Generation Efficiency and Feasibility in Neural Network Verification
>
>     We evaluated the UNSAT nodes and calculated the percentage that resulted in the generation of cuts. The table below provides a detailed breakdown:
>
>     | Dataset           | Avg. # Cuts Generated | Avg. UNSAT Nodes | Fraction of Cuts/UNSAT Nodes |
>     |-------------------|---------------------|------------------|------------------------------|
>     | cifar_cnn_a_adv   | 345.78              | 763.70           | 0.4528                       |
>     | cifar_cnn_a_adv4  | 86.33               | 2165.11          | 0.0399                       |
>     | cifar_cnn_a_mix   | 121.56              | 1766.77          | 0.0688                       |
>     | cifar_cnn_a_mix4  | 25.93               | 807.19           | 0.0321                       |
>     | cifar_cnn_b_adv   | 105.79              | 808.00           | 0.1309                       |
>     | cifar_cnn_b_adv4  | 106.14              | 1800.21          | 0.0590                       |
>     | mnist_cnn_a_adv   | 11.24               | 1351.67          | 0.0083                       |
>
>     Regarding the number of dropped assignments, this is related to the number of rounds of BaB (Branch and Bound) using our heuristic. Our setting has a drop ratio of 0.5 if the Lagrange factor of the neuron is 0, so the average number of dropped assignments will be less than 50%. This number decreases as BaB goes deeper and the domain becomes more refined.
>
>     Given this data, we agree with your suggestion that a cut selection procedure could be beneficial. We are implementing the new selection algorithm mentioned in **Q3** above and will report back.
>
> * Minor Issues
>
>     - Global set of cuts in strengthening: Thank you for noting this omission. We'll add explicit mention in the text that our bound computation during cut strengthening includes the global set of cuts, aligning with Algorithm 1.
>     - Introducing cut redundancy earlier: We agree this would improve readability. We'll revise Section 3.1 to briefly mention that the initial cuts are redundant until strengthened, providing context for Section 3.2.
>     - Multi-tree approach differences: We'll expand on how the trees differ in the multi-tree approach. Specifically, we'll clarify that in the first round, we select different neurons to start each tree, leading to diverse branching decisions. This will lead to different exploration paths.
>     - GPU usage in methods/baselines:
>         - CPU: nnenum, Marabou, Venus2.
>         - GPU: ERAN, OVAL, VeriNet, MN-BaB, PRIMA.
>     - Typos: we will fix typos and adjust the format according to the reviewer's feedback.

---

> > ### Comment · Reviewer_SS1K · 2024-08-08
> >
> > Thank you for the response. I have read the reviews and rebuttals and will keep my "Accept" rating. I do agree with other reviewers that this work is more incremental in nature, but I believe that the contribution is still sufficiently significant for acceptance. In my opinion, the ideas in this paper are interesting enough to publish and the presentation issues appropriately raised by other reviewers can be fixed for the final version. While other reviewers bring up topics that I had not considered, I believe the responses are satisfactory.
> >
> > A couple of minor comments:
> >
> > * *On binary search:* You can always limit the number of verification queries in your binary search. While I believe it is ok to leave it for future research, I suspect that it would work better than the one proposed in the paper. If you do move forward with future work in this direction (e.g. CDCL + your strengthening), I suggest considering this approach.
> >
> > * *Relationship to existing methods (based on other reviews):* There is some overlap with known methods from the SAT/MILP literature, but the methodological novelty here is the strengthening in the context of NN verification. During my review I did a quick search over the MILP literature to see if this strengthening approach already existed, because it is actually a rather simple method, and while there are similar approaches, I was surprised to see that it does not exist exactly in the way that is done here. From the SAT/CP literature, DPLL/CDCL/no-good learning is probably the closest one, but it is not quite the same.
> >
> >    A key here is that in NN verification we have these very fast, GPU-accelerated lower bounds (whereas MILP requires solving LPs). This opens the door to approaches like these which leverage these lower bounds, and it seems effective for verification, even if incrementally. This is also a reason why a custom B&B makes sense for NN verification. More speculatively, another reason why I think this works well is that in verification we focus on proving infeasibility, and in a way instances are expected to be tightly constrained. In particular, extracting cuts from the B&B tree is probably a good idea because ReLU LP relaxations in NN verification are very loose in deeper layers. While this can be naturally translated into other problems in MILP, I am skeptical it would be as effective in typical problems from the Operations Research community. My view of this paper is that it is an early step into incorporating CDCL-like ideas into the custom B&B framework for verification, much like ideas from MILP were incorporated in the past in the form of B&B and cuts, and I believe this is a positive step forward.
> >
> > Given that all improvements in the rebuttals are made (including the extra analysis that you made for this review, and especially better contextualization w.r.t. MILP, SAT, and other previous work as requested by both myself and iWFh), I support this paper for acceptance.

---

> > > ### Author Response · Authors · 2024-08-10
> > > **Thank you for the review**
> > >
> > > Thank you for your thoughtful comments and support for the paper.
> > >
> > > We appreciate your insights regarding the binary search method and its potential application in future work. Your feedback on the relationship to existing approaches, especially in the context of NN verification, is constructive. We will ensure that the final version addresses the presentation issues and provides better contextualization concerning MILP, SAT, and other related work.
> > >
> > > We're glad to have your support for acceptance and look forward to refining the paper accordingly.
> > >
> > > Best Regards,
> > >
> > > Anonymous Authors

---

### Author Rebuttal · Authors · 2024-08-07

Submission of figures of added experimental results

---

### Decision · Program_Chairs · 2024-09-25

**Decision:**

Accept (poster)

**Comment:**

The submission has received mixed ratings after extensive post-rebuttal discussions. One of the referees recommends rejection based on the weakness of the cuts that are generated by the proposed method, as well as incremental improvements over GCP-CROWN. However, the other referees recommend acceptance, and suggest acknowledging the aforementioned weaknesses in the revised version of the submission. The authors are advised to take into account the detailed reviews as well as the clarifications provided in the rebuttal when preparing the final manuscript.